# Using normative models pre-trained on cross-sectional data to evaluate intra-individual longitudinal changes in neuroimaging data

**Barbora Rehak Buckova[1,2,3], Charlotte Fraza[4], Rastislav Rehák[5,6], Marián Kolenič[3,7], Christian F Beckmann[4], Filip Španiel[3], Andre F Marquand[4]\*[†], Jaroslav Hlinka[1,3]\*[†]**

[1]Department of Complex Systems, Institute of Computer Science of the Czech Academy of Sciences, Prague, Czech Republic; [2]Department of Cybernetics, Czech Technical University in Prague, Prague, Czech Republic; [3]National Institute of Mental Health, Klecany, Czech Republic; [4]Donders Institute for Brain, Cognition and Behaviour, Nijmegen, Netherlands; [5]Max Planck Institute for Research on Collective Goods, Bonn, Germany; [6]University of Cologne, Köln, Germany; [7]Third faculty of medicine, Charles University, Prague, Czech Republic

**\*For correspondence:**
andre.marquand@donders.ru.
nl (AFM);
hlinka@cs.cas.cz (JH)

[†]These authors contributed equally to this work

## eLife Assessment

This paper addresses an **important** topic (normative trajectory modelling), seeking to provide a method aiming to accurately reflect the individual deviation of longitudinal/temporal change compared to the normal temporal change characterized based on a pre-trained population normative model. The evidence provided for the new methods is **solid**.

**Abstract** Longitudinal neuroimaging studies offer valuable insight into brain development, ageing, and disease progression over time. However, prevailing analytical approaches rooted in our understanding of population variation are primarily tailored for cross-sectional studies. To fully leverage the potential of longitudinal neuroimaging, we need methodologies that account for the complex interplay between population variation and individual dynamics. We extend the normative modelling framework, which evaluates an individual's position relative to population standards, to assess an individual's longitudinal *change* compared to the population's standard dynamics. Using normative models pre-trained on over 58,000 individuals, we introduce a quantitative metric termed '*z-diff*' score, which quantifies a temporal change in individuals compared to a population standard. This approach offers advantages in flexibility in dataset size and ease of implementation. We applied this framework to a longitudinal dataset of 98 patients with early-stage schizophrenia who underwent MRI examinations shortly after diagnosis and 1 year later. Compared to cross-sectional analyses, showing global thinning of grey matter at the first visit, our method revealed a significant normalisation of grey matter thickness in the frontal lobe over time—an effect undetected by traditional longitudinal methods. Overall, our framework presents a flexible and effective methodology for analysing longitudinal neuroimaging data, providing insights into the progression of a disease that would otherwise be missed when using more traditional approaches.

## Introduction

Longitudinal neuroimaging studies provide a unique opportunity to gain insight into the temporal dynamics of a disease, over and above the insights offered by cross-sectional studies. Consequently, it is crucial to have tools to effectively analyse them whilst also making use of more widely available cross-sectional data to refine inferences. Therefore, in this manuscript, we develop a novel method for evaluating longitudinal *changes* in a subject's neuroimaging data, building upon an existing normative modelling framework originally designed to assess a subject's *position* within a population. This adaptation allows us to track individual changes over time, providing a more dynamic understanding of neuroanatomical variations.

Normative modelling is a promising technique for modelling population variation in neuroimaging data (*Marquand et al., 2016*; *Bethlehem et al., 2022*). This framework models each image-derived phenotype (IDP) (e.g. voxel intensity, regional thickness, or regional volume) independently as a function of demographic or clinical variables (e.g. age, sex, scanning site) in a large healthy population. Subjects are subsequently compared to the normative model characterising the healthy population, which enables us to evaluate the position of each individual, rather than just compare group differences between patients and controls (*Fraza et al., 2021*; *Habes et al., 2021*). Application of these models has already provided valuable insights into the individual neuroanatomy of various diseases, such as Alzheimer's, schizophrenia, autism, and other neurological and mental disorders (*Pinaya et al., 2021*; *Wolfers et al., 2021*; *Zabihi et al., 2019*; *Marquand et al., 2019*).

Longitudinal data are conceptually well suited to extend standard normative modelling since they analyse individual trajectories over time. If adjusted appropriately, normative models could not only improve predictive accuracy but also identify patterns of temporal change, thereby enhancing our understanding of the disease.

However, despite their potential, longitudinal normative models have not yet been systematically explored (*Di Biase et al., 2023*; *Bethlehem et al., 2022*). Indeed, virtually all large-scale normative models released to date are estimated on cross-sectional data (*Rutherford et al., 2022*; *Bethlehem et al., 2022*) and a recent report (*Di Biase et al., 2023*) has provided empirical data to suggest that such cross-sectional models may underestimate the variance in longitudinal data (*Di Biase et al., 2023*). However, from a theoretical perspective, it is very important to recognise that cross-sectional models describe group-level population variation across the lifespan, where such group-level centiles are interpolated smoothly across time. It is well known in the pediatric growth-charting literature (e.g. *Cole, 2012*) that centiles in such cross-sectional models do not necessarily correspond to individual-level trajectories, rather it is possible that individuals cross multiple centiles as they proceed through development, even in the absence of pathology. Crucially, classical growth charts and current normative brain charts provide no information about how frequent such centile crossings are in general. In other words, they provide a *trajectory of distributions*, not a *distribution over trajectories*. There are different approaches to tackle this problem in the growth charting literature, including the estimation of 'thrive lines' that map centiles of constant velocity across the lifespan and can be used to declare 'failure to thrive' at the individual level (e.g. see *Cole, 2012*, for details). Unfortunately, this approach requires densely sampled longitudinal neuroimaging data to estimate growth velocity, that are not available across the human lifespan at present. Therefore, in this work, we adopt a different approach based on estimates of the uncertainty in the centile estimates themselves together with the uncertainty with which a point is measured (e.g. bounded by the test-retest reliability, noise, etc.). By accounting for such variability, this provides a statistic to determine whether a centile crossing is large enough to be statistically different from the base level within the population.

We stress that our aim is not to build a longitudinal normative model per se. Considering the much greater availability of cross-sectional data relative to longitudinal data, we instead leverage existing models constructed from densely sampled cross-sectional populations and provide methods for applying these to longitudinal cohorts. We argue that although these models lack explicit intra-subject dynamics, they contain sufficient information to enable precise assessments of changes over time. Nevertheless, the inclusion of longitudinal data into existing models largely estimated from cross-sectional data is also an important goal and can be approached with hierarchical models (*Kia et al., 2022*); however, we do not tackle this problem here.

We derive a novel set of difference ('*z-diff*') scores for statistical evaluation of longitudinal change between two measurements (the 'diff' in the name refers to the temporal difference that we are

evaluating as opposed to a one-time position evaluated by the simple $z$-score). We utilise the warped Bayesian linear regression normative model (*Fraza et al., 2021*) as a basis for our work. Training these models requires significant amounts of data and computational resources, limiting their use for smaller research groups. However, the availability of pre-trained models has made them more accessible to researchers from a wider range of backgrounds, as reported by *Rutherford et al., 2022*. We present a comprehensive theoretical analysis of our method, followed by numerical simulations and a practical application to an in-house longitudinal dataset of 98 patients in the early stages of schizophrenia who underwent fMRI examinations shortly after being diagnosed and 1 year after.

## Methods
### Model formulation
#### Original model for cross-sectional data
Here, we briefly present the original normative model (*Fraza et al., 2021*), developed to be trained and used on cross-sectional data. In the following subsection, we take this model pre-trained on a large cross-sectional dataset and extend it so that it can be used on longitudinal data.

The original model (*Fraza et al., 2021*) is pre-trained on a cross-sectional dataset $\mathbf{Y} = (y_{nd}) \in \mathbb{R}^{N \times D}$, $\mathbf{X} = (x_{nm}) \in \mathbb{R}^{N \times M}$ of $N$ subjects, for whom we observe $D$ IDPs and $M$ covariates (e.g. age or sex). Thus, $y_{nd}$ is the $d$th IDP of the $n$th subject and $x_{nm}$ is the $m$th covariate of the $n$th subject.

Since each IDP is treated separately, we focus on a fixed IDP $d$ and drop this index for ease of exposition. To simplify notation, we denote $\mathbf{y} = (y_1, \ldots, y_N)^T$ the column of observations of this fixed IDP across subjects. The observations are assumed to be independent (across $n$ subjects). To model the relationship between IDP $y_n$ and covariates $\mathbf{x}_n = (x_{n1}, \ldots, x_{nM})^T$, we want to exploit a normal linear regression model described in *Fraza et al., 2021*; *Rutherford et al., 2022*. However, we make a couple of adjustments first:

- To accommodate non-Gaussian errors in the original space of dependent variables, we transform the original variable $y_n$ by a warping function $\varphi(y_n)$, which is parametrised by hyperparameters $\gamma$ (see Section 2.3 in *Fraza et al., 2021*, for details).
- To capture non-linear relationships, we use a B-spline basis expansion of the original independent variables $\mathbf{x}_n$ (see Section 2.3 in *Fraza et al., 2021*, for details). To accommodate site-level effects, we append it with site dummies. We denote the resulting transformation of $\mathbf{x}_n$ as $\phi(\mathbf{x}_n) \in \mathbb{R}^K$.

We also treat the variance of measurements as a hyper-parameter and we denote it by $\sigma^2$. Thus, we model the distribution of the transformed IDP $\varphi(y_n)$ conditional on covariates $\mathbf{x}_n$, vector of parameters $\mathbf{w}$, and hyper-parameters $\sigma^2$ and $\gamma$ as follows:

$$\varphi(y_n) = \mathbf{w}^T \phi(\mathbf{x}_n) + \varepsilon_n, \ \varepsilon_n \sim \mathcal{N}(0, \sigma^2), \tag{1}$$

where $\varepsilon_n$ are independent from $x_n$ and across $n$. We further denote the design matrix $\mathbf{\Phi} = (\phi(\mathbf{x}_n)_k) \in \mathbb{R}^{N \times K}$ ($\phi(\mathbf{x}_n)_k$ is the $k$the element of vector $\phi(\mathbf{x}_n)$).

The estimation of parameters $\mathbf{w}$ is performed by empirical Bayesian methods. In particular, prior about $\mathbf{w}$

$$\mathcal{N}(0, \omega^2 \mathbf{I}) \tag{2}$$

is combined with the likelihood function to derive the posterior

$$\mathbf{w} | \mathbf{y}, \mathbf{\Phi}; \omega^2, \sigma^2, \gamma \sim \mathcal{N}(\bar{\mathbf{w}}, \mathbf{A}^{-1}), \tag{3}$$

$$\mathbf{A} = \sigma^{-2} \mathbf{\Phi}^T \mathbf{\Phi} + \omega^{-2} \mathbf{I}, \tag{4}$$

$$\bar{\mathbf{w}} = \sigma^{-2} \mathbf{A}^{-1} \mathbf{\Phi}^T \mathbf{y} \tag{5}$$

The hyper-parameters $\omega^2$, $\sigma^2$, $\gamma$ are estimated by maximising the warped marginal log-likelihood. The predictive distribution of $\varphi(y)$ for a subject with $x$ is

$$\mathcal{N} \left( \bar{\mathbf{w}}^T \phi(\mathbf{x}), \phi(x)^T A^{-1} \phi(x) + \sigma^2 \right). \tag{6}$$

Hence, the $z$-score characterising the position of this subject within population is

$$z = \frac{\varphi(y) - \bar{\mathbf{w}}^T \phi(\mathbf{x})}{\sqrt{\phi(\mathbf{x})^T \mathbf{A}^{-1} \phi(\mathbf{x}) + \sigma^2}}, \tag{7}$$

where $\varphi(y)$ is the realised warped observation of IDP $d$ for this subject. This score captures how surprising is the actual observation $\varphi(y)$ relative to what one would expect for an average subject with the same characteristics $\bar{\mathbf{w}}^T \phi(x)$, and this deviation has to be compared with (normalised by) the variability stemming from the natural variability in the data ($\sigma^2$) and the modelling uncertainty ($\phi(\mathbf{x})^T \mathbf{A}^{-1} \phi(\mathbf{x})$).

In this form, the original models were fit on a large dataset consisting of 58,836 participants scanned across 82 sites. Specifically, cortical thickness and subcortical volumes were modelled, and the models were validated against a subset of 24,000 participants, the quality of which was checked manually (*Rutherford et al., 2022*).

Note that *Equation 6* and *Equation 7* implicitly evaluate only (potentially new) subjects measured at sites already present in the original dataset $\mathbf{y}$, $\boldsymbol{\Phi}$. If we want to evaluate subjects measured at a new site, we will have to run an adaptation procedure to account for its effect. This adaptation procedure is described and readily accessible online in *Rutherford et al., 2022*. In short, a sample of a reference (healthy) cohort measured on the same scanner as the population of interest is needed to accommodate a site-specific effect.

In the following section, we develop a procedure that allows us to extend the original cross-sectional framework pre-trained on dataset $\mathbf{y}$, $\boldsymbol{\Phi}$ to evaluate a new longitudinal dataset for assessment of temporal changes.

## Adaptation to longitudinal data

We adapt the original cross-sectional normative modelling framework (*Fraza et al., 2021*) (reviewed in the previous section) to the evaluation of intra-subject longitudinal changes. Specifically, we design a score for a longitudinal change between visits (further referred to as *z-diff* score), based on which we can assess temporal changes in regional brain thickness and potentially detect any unusually pronounced deviations from normative trajectories.

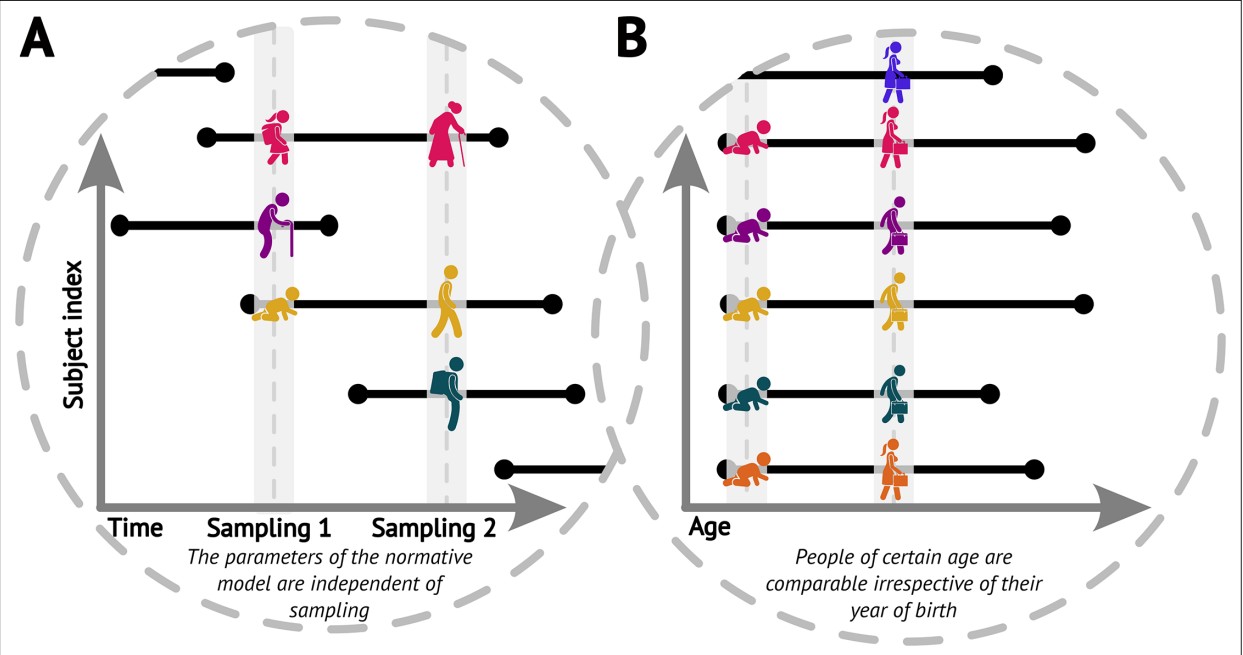

**Figure 1.** Visualisations of the core assumptions of normative modelling. (**A**) The parameters of the fitted normative model are independent of the time of sampling. (**B**) People of the same age are comparable irrespective of their year of birth (datasets sampled at different times can be combined).

We start by noticing that the original cross-sectional normative modelling framework (*Fraza et al., 2021*) features an implicit assumption that pertains to the longitudinal view. Specifically, it assumes that had we randomly sampled the population at a different time (e.g. 5 years sooner or later), we would have gotten equivalent picture about the 'norm' (up to randomness of the sampling, *Figure 1A*). In other words, the parameters of the normative model would be the same irrespective of the time we sampled the population, including the case in which we would sample the same people again, just later (while appropriately compensating for the resulting under-representation of younger ages). We further assume comparability of people of any given age irrespective of their birth time (i.e. we assume independence of birth dates and trajectories, *Figure 1B*). Together, these assumptions imply a form of stationarity (formally discussed in the next paragraph). These are indispensable assumptions for the practical usefulness of normative modelling, albeit one can see that in the real world they are not fulfilled perfectly, e.g., due to evolutionary dynamics, the ever-changing environment, or any changes in the distribution of those demographic variables that are not explicitly accounted for in the normative model.

Formally, we work with the process $\{(y_{n,t}, \mathbf{x}_{n,t})\}$, where $n$ indexes a subject and $t$ indexes age (to avoid technicalities, we assume discrete time). The minimal requirement imposed implicitly by the above assumptions is $\varepsilon_t \sim \mathcal{N}(0, \sigma^2)$ for every age $t$. We further restrict our focus to the class of stationary Gaussian processes $\varepsilon$, i.e., processes such that $\varepsilon_{t_1}, \varepsilon_{t_2}, \ldots, \varepsilon_{t_k}$ are jointly normal for any finite set of ages $t_1, t_2, \ldots, t_k$ and their joint distribution is invariant to any admissible time shift $t_1 + s, t_2 + s, \ldots, t_k + s$. Furthermore, we focus on a specific process in this section for ease of exposition, and we discuss the general case in the next section.

The specific process we focus on in this section reflects the assumption that a healthy subject does not deviate substantially from their position within the population as they get older (*Marquand et al., 2019*); the observed position change between the visits stems from observation noise (due to technical or physiological factors) and is therefore constrained by the test-retest reliability of the measurement. Formally,

$$\varepsilon_{n,t} = \eta_n + \xi_{n,t},$$

where $\eta$ is a subject-specific time-independent factor independent of the iid noise process $\xi$. Note that this does not imply that a healthy subject does not change over time, but rather that the change follows approximately the population centile at which the individual is placed. We generalise our method to other stationary Gaussian processes in the next section.

According to this model, the $i$th visit of a healthy subject with given covariates $\mathbf{x}^{(i)}$ is generated by

$$\varphi(y^{(i)}) = \mathbf{w}^T \phi(\mathbf{x}^{(i)}) + \eta + \xi^{(i)}$$
$$\eta \sim \mathcal{N}(0, \sigma_\eta^2)$$
$$\xi^{(i)} \sim \mathcal{N}(0, \sigma_\xi^2) \tag{8}$$
$$\sigma^2 = \sigma_\eta^2 + \sigma_\xi^2$$

where $\eta$, $\xi^{(i)}$, and $\mathbf{x}^{(i)}$ are mutually independent for a given $i$, and the measurement errors $\xi^{(i)}$ and $\xi^{(j)}$ are independent across visits $i \neq j$. Note that we dropped the subject-specific index $n$ (and subsumed the age $t_n^{(i)}$ in the visit index $i$). This should remind the reader that the goal is just to evaluate longitudinal change of a given subject from our new longitudinal data and not to re-estimate the parameters with these additional data. Nevertheless, to properly adapt the cross-sectional model, we will need to estimate one new parameter stemming from the further structure we impose on $\varepsilon$.

In our longitudinal data, we are interested in the change for a given individual across two visits. According to *Equation 8*, the difference in the transformed IDP between visits 1 and 2, $\varphi(y^{(2)}) - \varphi(y^{(1)})$, for a subject with covariates $\mathbf{x}^{(1)}$ and $\mathbf{x}^{(2)}$ is given by

$$\varphi(y^{(2)}) - \varphi(y^{(1)}) = \mathbf{w}^T[\phi(\mathbf{x}^{(2)}) - \phi(\mathbf{x}^{(1)})] + \xi^{(2)} - \xi^{(1)} \tag{9}$$

with $\xi^{(2)} - \xi^{(1)} \sim \mathcal{N}(0, 2\sigma_\xi^2)$. We use the posterior distribution of $\mathbf{w}$ with hyper-parameters $\omega^2, \sigma^2, \gamma$ estimated on the original cross-sectional dataset $\mathbf{y}, \boldsymbol{\Phi}$ (the estimates are available at https://github.com/predictive-clinical-neuroscience/braincharts; *Rutherford and Marquand, 2024*). Therefore, the

posterior predictive distribution for the difference $\varphi(y^{(2)}) - \varphi(y^{(1)})$ for our subject is (for more detailed derivation, please refer to the Appendix 1)

$$\mathcal{N}\left(\bar{\mathbf{w}}^T[\phi(\mathbf{x}^{(2)}) - \phi(\mathbf{x}^{(1)})], [\phi(\mathbf{x}^{(2)}) - \phi(\mathbf{x}^{(1)})]^T\mathbf{A}^{-1}[\phi(\mathbf{x}^{(2)}) - \phi(\mathbf{x}^{(1)})] + 2\sigma_\xi^2\right). \tag{10}$$

Hence, the $z$-score for the difference in the transformed IDP between visits 1 and 2 is

$$z\text{-}diff = \frac{[\varphi(y^{(2)}) - \varphi(y^{(1)})] - \bar{\mathbf{w}}^T[\phi(\mathbf{x}^{(2)}) - \phi(\mathbf{x}^{(1)})]}{\sqrt{[\phi(\mathbf{x}^{(2)}) - \phi(\mathbf{x}^{(1)})]^T\mathbf{A}^{-1}[\phi(\mathbf{x}^{(2)}) - \phi(\mathbf{x}^{(1)})] + 2\sigma_\xi^2}}, \tag{11}$$

where $\varphi(y^{(2)}) - \varphi(y^{(1)})$ is the realised temporal change in the warped observations of the IDP for this subject. Since this $z$-diff score is standard normal for the population of healthy controls, any large deviations may be used to detect unusual temporal changes.

The primary role of adaptation of the (pre-trained) cross-sectional model to (new) longitudinal data is to account for the measurement noise variance $\sigma_\xi^2$, thus, taking care of the atemporal source of variability $\eta$. In other words, having an estimate of $\sigma_\xi^2$ in hand helps us to use the proper scaling. To arrive at an estimator of $\sigma_\xi^2$, notice that from the posterior predictive distribution (**Equation 10**), we have (denoting the set of conditionals $\Omega = \{\mathbf{x}^{(1)}, \mathbf{x}^{(2)}; \mathbf{y}, \mathbf{\Phi}; \omega^2, \sigma^2, \boldsymbol{\gamma}\}$)

$$\mathrm{E}\left[\left(\varphi(y^{(2)}) - \varphi(y^{(1)}) - \mathrm{E}[\varphi(y^{(2)}) - \varphi(y^{(1)})|\Omega]\right)^2\Big|\Omega\right]$$
$$= [\phi(\mathbf{x}^{(2)}) - \phi(\mathbf{x}^{(1)})]^T\mathbf{A}^{-1}[\phi(\mathbf{x}^{(2)}) - \phi(\mathbf{x}^{(1)})] + 2\sigma_\xi^2. \tag{12}$$

Hence, by the law of iterated expectations (to integrate out $\mathbf{x}^{(1)}$ and $\mathbf{x}^{(2)}$), we obtain

$$\mathrm{E}\left[\left(\varphi(y^{(2)}) - \varphi(y^{(1)}) - \bar{\mathbf{w}}^T[\phi(\mathbf{x}^{(2)}) - \phi(\mathbf{x}^{(1)})]\right)^2\right.$$
$$\left. - [\phi(\mathbf{x}^{(2)}) - \phi(\mathbf{x}^{(1)})]^T\mathbf{A}^{-1}[\phi(\mathbf{x}^{(2)}) - \phi(\mathbf{x}^{(1)})]\Big|\mathbf{y}, \mathbf{\Phi}; \omega^2, \sigma^2, \boldsymbol{\gamma}\right] = 2\sigma_\xi^2. \tag{13}$$

Therefore, we estimate $2\sigma_\xi^2$ by the sample analogue of the left-hand side in **Equation 13**. Specifically, we devote a subsample $C$ of the controls from our (new) longitudinal data just to this estimation (i.e. the subjects from $C$ will not be used in the evaluation) and we compute

$$\widehat{2\sigma_\xi^2} = \frac{1}{|C|}\sum_{k\epsilon C}\left[\left(\varphi(y_k^{(2)}) - \varphi(y_k^{(1)}) - \bar{\mathbf{w}}^T[\phi(\mathbf{x}_k^{(2)}) - \phi(\mathbf{x}_k^{(1)})]\right)^2\right.$$
$$\left. - [\phi(\mathbf{x}_k^{(2)}) - \phi(\mathbf{x}_k^{(1)})]^T\mathbf{A}^{-1}[\phi(\mathbf{x}_k^{(2)}) - \phi(\mathbf{x}_k^{(1)})]\right], \tag{14}$$

where $|C|$ denotes the number of subjects in subsample $C$.

Another useful feature of longitudinal data is that $[\phi(\mathbf{x}_k^{(2)}) - \phi(\mathbf{x}_k^{(1)})]$ is negligible (especially with stable covariates, like sex and age). Sex (typically) does not change across the two visits and age relatively little (in our target application) with respect to the full span of ageing. Consequently, $[\phi(\mathbf{x}_k^{(2)}) - \phi(\mathbf{x}_k^{(1)})]^T\mathbf{A}^{-1}[\phi(\mathbf{x}_k^{(2)}) - \phi(\mathbf{x}_k^{(1)})]$ in **Equation 14** is negligible in adult cohorts but must be treated with caution in developmental or ageing groups. Finally, it is apparent from **Equation 4** that $\mathbf{A}$ scales with the number of subjects, and its inverse will be negligible for substantial training datasets, such as the one that was used for pre-training.

To conclude this subsection, we caution against the naïve use of the difference of the simple $z$-scores

$$\frac{\varphi(y^{(2)}) - \bar{\mathbf{w}}^T\phi(\mathbf{x}^{(2)})}{\sqrt{\phi(\mathbf{x}^{(2)})^T\mathbf{A}^{-1}\phi(\mathbf{x}^{(2)}) + \sigma^2}} - \frac{\varphi(y^{(1)}) - \bar{\mathbf{w}}^T\phi(\mathbf{x}^{(1)})}{\sqrt{\phi(\mathbf{x}^{(1)})^T\mathbf{A}^{-1}\phi(\mathbf{x}^{(1)}) + \sigma^2}} \tag{15}$$

instead of the $z$-diff score to evaluate the longitudinal change. The problem with such an approach is apparent by comparing it to the $z$-diff score (**Equation 11**); it does not properly account for the modelling uncertainty (instead of using the combined term $[\phi(\mathbf{x}^{(2)}) - \phi(\mathbf{x}^{(1)})]^T\mathbf{A}^{-1}[\phi(\mathbf{x}^{(2)}) - \phi(\mathbf{x}^{(1)})]$ to scale the difference of the numerators in **Equation 15**, it scales the individual terms of the difference by

their individual model uncertainty). More importantly, even if the modelling uncertainty is negligible, *Equation 15* does not properly scale the difference of the 'residuals' because it incorrectly includes the common source of subject-level variability $\eta$ (it uses $\sigma_\eta^2 + \sigma_\xi^2$ instead of $2\sigma_\xi^2$), as we later demonstrate in the simulation part of the study.

## More general dynamics

The model we introduced in the previous section is an intuitive extension of the original model introduced in the 'Model formulation' section. However, the model operates with a seemingly strong (although reasonable) assumption that healthy subjects inherently follow their centiles. Due to the lack of large longitudinal data testing this assumption, in this section, we investigate the generalisation to other stationary Gaussian processes to illustrate the robustness of our method. As an example, we are able to deal with a stationary Gaussian AR(1) process $\varepsilon_t = \zeta\varepsilon_{t-1} + \xi_t$ with $|\zeta| < 1$, $\xi$ iid $\mathcal{N}(0, (1 - \zeta^2)\sigma^2)$, and $\varepsilon_0 \sim \mathcal{N}(0, \sigma^2)$.

Importantly, our framework evaluates change only between two visits. Hence, we do not need to consider the full specification of the process $\varepsilon$, but only the time dependence between the two visits that can arise under it. Formally, since we are in the class of stationary Gaussian processes, we only need to consider the autocorrelation between $\varepsilon^{(1)}$ and $\varepsilon^{(2)}$ $\rho \in [-1, 1]$. Just as an example, the stationary Gaussian AR(1) process introduced above would produce autocorrelation $\rho = \zeta^{T_2 - T_1}$, where $T_2 - T_1$ is the time between the two visits.

Considering this more general class of processes, this amounts to $\varepsilon^{(2)} - \varepsilon^{(1)} \sim \mathcal{N}(0, 2\sigma^2(1 - \rho))$. Going through the same derivations as before, we obtain the score for the evaluation of longitudinal change

$$z\text{-diff} = \frac{[\varphi(y^{(2)}) - \varphi(y^{(1)})] - \bar{\mathbf{w}}^T[\phi(\mathbf{x}^{(2)}) - \phi(\mathbf{x}^{(1)})]}{\sqrt{[\phi(\mathbf{x}^{(2)}) - \phi(\mathbf{x}^{(1)})]^T\mathbf{A}^{-1}[\phi(\mathbf{x}^{(2)}) - \phi(\mathbf{x}^{(1)})] + 2\sigma^2(1 - \rho)}} \tag{16}$$

and the estimator

$$\widehat{2\sigma^2(1 - \rho)} = \frac{1}{|C|}\sum_{k \in C}\Big[\left(\varphi(y_k^{(2)}) - \varphi(y_k^{(1)}) - \bar{\mathbf{w}}^T[\phi(\mathbf{x}_k^{(2)}) - \phi(\mathbf{x}_k^{(1)})]\right)^2 \\ - [\phi(\mathbf{x}_k^{(2)}) - \phi(\mathbf{x}_k^{(1)})]^T\mathbf{A}^{-1}[\phi(\mathbf{x}_k^{(2)}) - \phi(\mathbf{x}_k^{(1)})]\Big]. \tag{17}$$

These take the same form as in the specific case considered in the previous section. Hence, the method developed in the previous section does not depend on that particular assumption about the process $\varepsilon$ and will still yield valid inferences even if the seemingly strong assumption of centile tracking is violated. In either case, we need one more free parameter to properly account for the potential non-iid dynamics of $\varepsilon$ in the previous section, $\rho$ in this section. The only substantial difference is that while $2\sigma^2(1 - \rho)$ can be larger than $2\sigma^2$ (for $\rho < 0$), the process from the previous section leads to $2\sigma_\xi^2$ with values only lower than $2\sigma^2$. This could provide a test of the assumption about the process from the previous section: if our estimate $\widehat{2\sigma_\xi^2}$ is larger than the cross-sectional estimate $\widehat{2\sigma^2}$, then, the assumption about $\varepsilon$ in the previous section is not justified.

## Simulation study

To formally evaluate the performance of the proposed method in making accurate inferences about the longitudinal changes, we conduct a simulation study. We imagine a practitioner who would use some lower and upper thresholds for the *z-diff* score to detect unusual change. It is natural to choose the probability of dubbing a healthy control as unusual $\theta \in [0, 1]$, and use the $\frac{\theta}{2}$ and $1 - \frac{\theta}{2}$ quantiles of the standard normal distribution as the thresholds (denote them $q_{\frac{\theta}{2}}$ and $q_{1-\frac{\theta}{2}}$, respectively). A subject with *z-diff* $< q_{\frac{\theta}{2}}$ or $> q_{1-\frac{\theta}{2}}$ is thus flagged as someone with unusual change. We would like to know how successfully this classification detects true changes, i.e., how often it detects a patient with a disrupted trajectory. We capture the disruption by a process $\delta$, i.e., the trajectory of a patient in our model would be $\varphi(y)$. We treat the realised change in the disruption between the two visits $\Delta := \delta^{(2)} - \delta^{(1)}$ as a fixed number to be detected.

For the simulation, we fix $\theta = 0.05$. For each combination of $\Delta \in [-4, 4]$ and $\rho \in (-1, 1)$, we generate a large number of patients with various age and gender, disruption $\Delta$, and $(\varepsilon^{(2)}, \varepsilon^{(1)})$ from the bivariate normal distribution

$$\mathcal{N}\left(\begin{bmatrix} 0 \\ 0 \end{bmatrix}, \begin{bmatrix} \sigma^2 & \rho\sigma^2 \\ \rho\sigma^2 & \sigma^2 \end{bmatrix}\right). \tag{18}$$

Specifically, we produce $\varphi(y^{(1)}) = \bar{\mathbf{w}}^T\phi(\mathbf{x}^{(1)}) + \varepsilon^{(1)}$, $\varphi(y^{(2)}) = \bar{\mathbf{w}}^T\phi(\mathbf{x}^{(2)}) + \varepsilon^{(2)} + \Delta$ (the remaining parameters are the cross-sectional estimates). For each patient, we calculate the *z-diff* score and we look at the fraction of patients with a *z-diff* score surpassing the thresholds. Additionally, we compare the results to the naïve approach based on evaluating the difference of the *z*-scores between the two visits, while using the same thresholds (the quantiles of the standard normal distribution). The resulting simulation is depicted in *Figure 2*.

Two intuitive properties arise from this simulation: larger disruptions are easier to detect; positive autocorrelation in $\varepsilon$ makes it easier to detect the disruptions, while negative autocorrelation makes it harder. Strong positive autocorrelation reflects a strong common component in $\varepsilon$ across the two visits, which cancels out through subtracting the two visits, while strong negative autocorrelation indicates strong switching in $\varepsilon$ across the two visits, which can be easily confounded with the true disruption $\Delta$. Finally, if the true process for $\varepsilon$ is as assumed in *Equation 8* (stable component plus noise), then, higher noise $\sigma_\xi^2$ corresponds to lower (but positive) $\rho$. Intuitively, we can see that more noise makes the detection of true disruptions more difficult.

The results of the simulation also clearly caution against the use of the naïve *z*-score subtraction (*Figure 2*). First, the *z-diff* method maintains a consistent false-positive rate (when no disruption is present) of 5%, unlike the *z*-score subtraction, where the false-positive rate changes with the autocorrelation.

Second, in terms of detection power the *z-diff* method outperforms the simple *z*-score subtraction for high autocorrelation, in particular when the autocorrelation is above 0.5, with the difference in the performance being more pronounced with rising autocorrelation. This is particularly relevant in practice—our real-world data imply very high values of $\rho$ (average $\hat{\rho}$ across all IDPs is 0.9; *Appendix 2—figure 1*).

Finally, the seemingly better performance of the *z*-score subtraction under autocorrelation below 0.5 is caused only by the subtraction method's general tendency to label cases as 'suspicious' in the absence of change, e.g., by the increased false-positive rate (*Figure 2*, right). This leads to a small steady 'improvement' across the space of disruptions. See more detailed simulation results in *Appendix _3—figure 1*.

Let us revisit the earlier example of a practitioner using these methods to identify unusual changes in patients. In most real-world scenarios, the majority of people are healthy, with only a few experiencing pronounced changes that might indicate illness. If the detection method does not properly control the false-positive rate, many healthy individuals could be mistakenly flagged as needing further investigation. This could lead to unnecessary stress, costly follow-ups, or even painful procedures for those individuals. The naïve subtraction of *z*-scores is problematic in this regard. It often misclassifies healthy individuals, particularly when the correlation between measurements (autocorrelation) is low. This inconsistency makes it unreliable in practice. In contrast, the *z-diff* method consistently maintains a predictable false-positive rate while also improving the ability to detect true changes (when measurements are highly positively correlated). This balance ensures that more patients with real disruptions are identified while minimising unnecessary interventions for healthy individuals.

## Implementation

To implement the method (*Figure 3*), we used the PCN toolkit. The exact steps of the analysis with detailed explanations are available in the online tutorial at `PCNtoolkit-demo` (https://github.com/predictive-clinical-neuroscience/PCNtoolkit-demo) in the tutorials section.

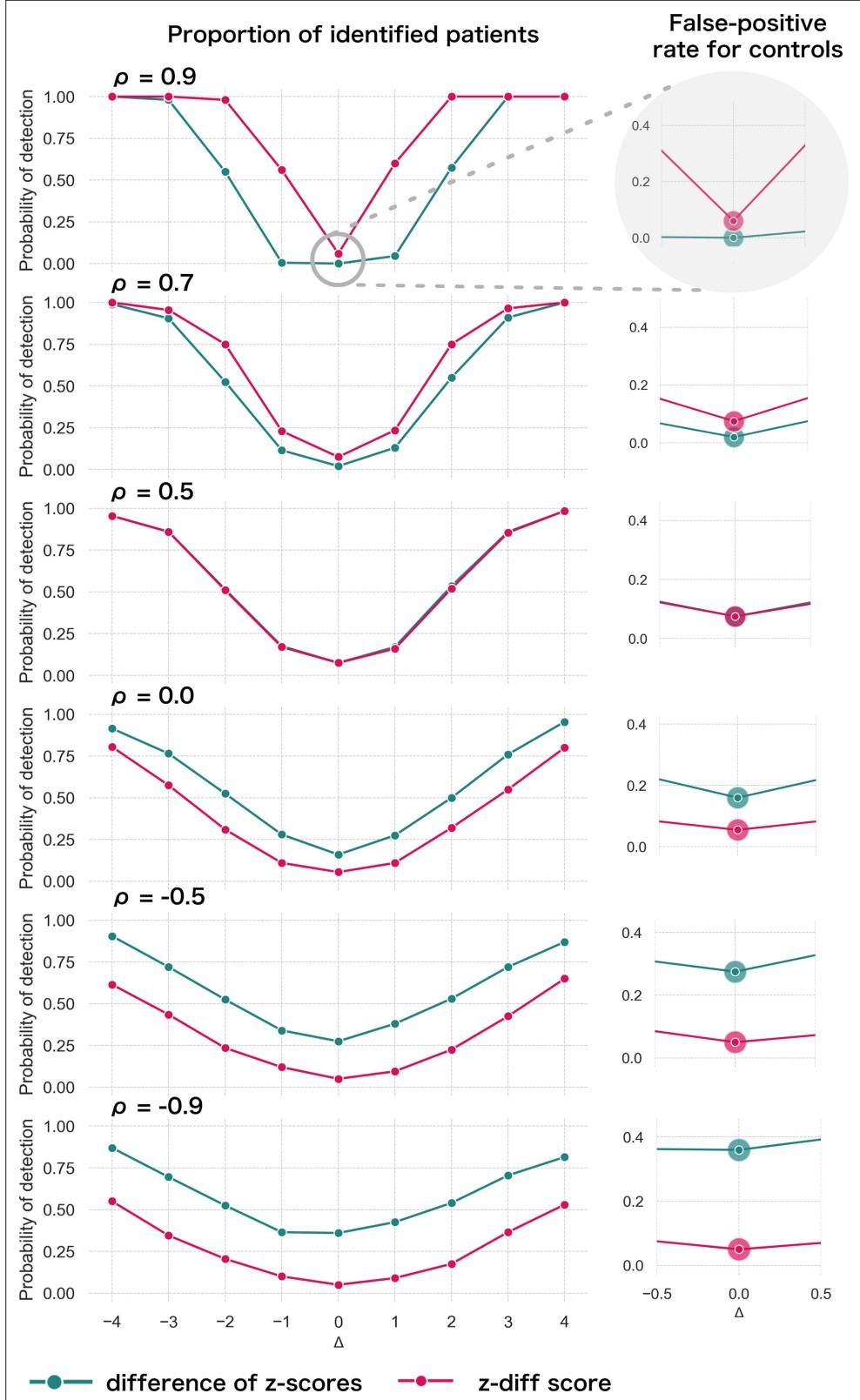

**Figure 2.** Simulated detection rate of a true disruption $\Delta$ for various values of autocorrelation $\rho$ (individual subplots) comparing the performance of our *z-diff* method against the naïve subtraction of *z*-scores. The right column highlights the false-positive rate across various degrees of autocorrelation for the two approaches. We use $\sigma^2 = 1$ and $\theta = 0.05$.

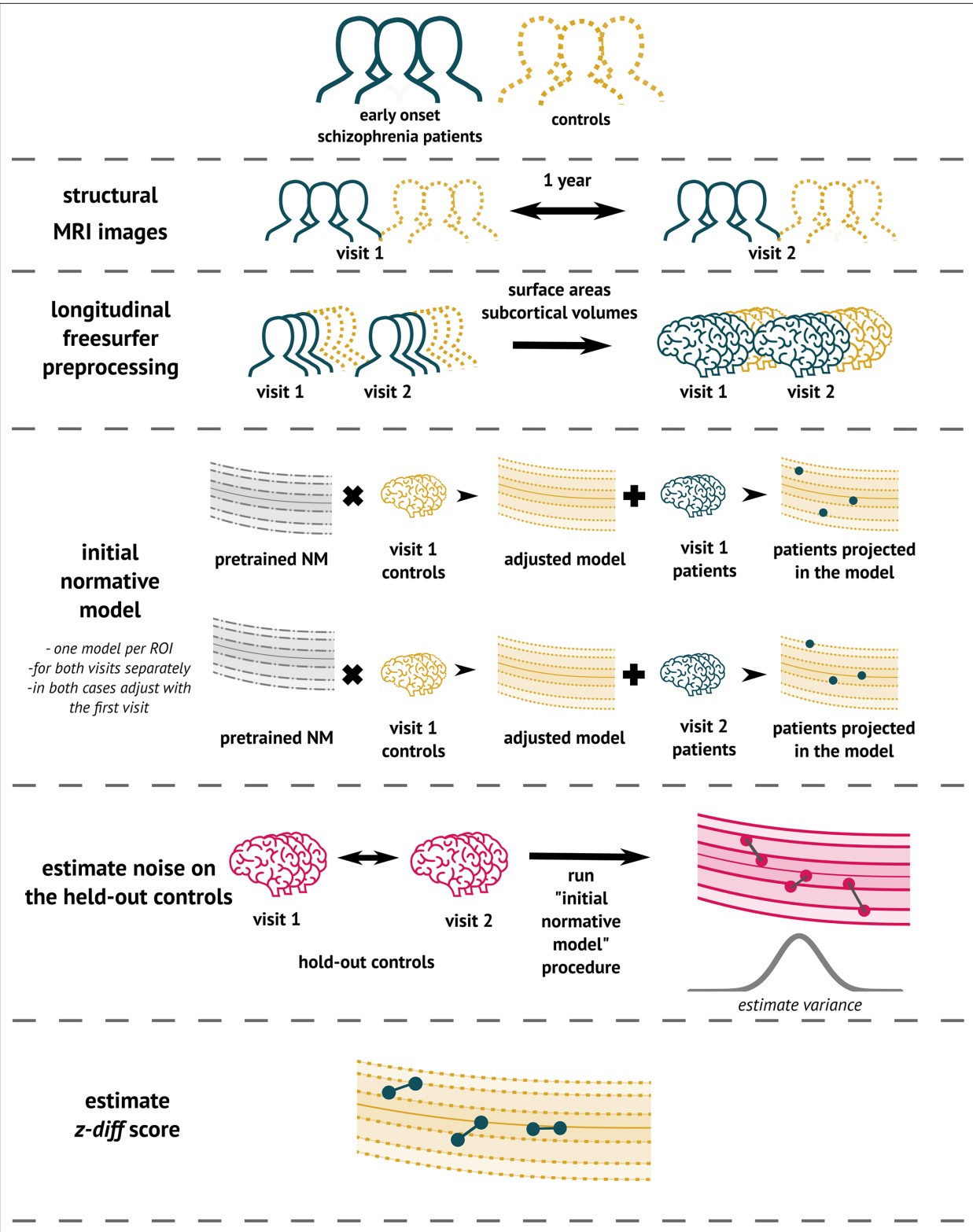

**Figure 3.** The overview of the analytical pipeline for our schizophrenia patients: First, data are preprocessed using FreeSurfer's longitudinal pipeline. Subsequently, the pre-trained models are adjusted to a local sample of healthy controls. The site-specific measurement noise variance $\sigma_\xi^2$ in healthy subjects is estimated using held-out controls, and finally, the *z-diff* score is computed.

**Table 1.** Clinical description of the dataset after quality control.

| | Patients | Controls |
|---|---|---|
| *N* (% females) | 98 (39%) | 67 (63%) |
| Age, median (min, max), years | 27 (18, 46) | 29 (18, 54) |
| Interval between visits, median (min, max), years | 1.1 (0.9,2.7) | 1.2 (0.9, 3) |
| *Diagnosis (only for patients)* | | |
| Schizophrenia | 53 | |
| Brief psychotic disorder | 45 | |
| Length of disease, median (min, max), months | 4 (1,21) | |
| *Clinical scales (only for patients)* | Visit 1 | Visit 2 |
| PANSS sum, median (min, max) | 53 (30, 94) | 44 (30, 84) |
| PANSS positive symptoms, median (min, max) | 11 (7, 21) | 8 (7, 26) |
| PANSS negative symptoms, median (min, max) | 14.5 (7, 30) | 11.5 (7, 24) |
| GAF, median (min, max) | 70 (25, 100) | 80.5 (40, 98) |

## Data

### Early stages of schizophrenia patients

The clinical data used for the analysis were part of the Early Stages of Schizophrenia study (*Spaniel et al., 2016*). We analysed data from 98 patients in the early stages of schizophrenia (38 females) and 67 controls (42 females) (*Table 1*). The inclusion criteria were as follows: The subjects were over 18 years of age and undergoing their first psychiatric hospitalisation. They were diagnosed with schizophrenia, or acute and transient psychotic disorders, and suffered from untreated psychosis for less than 24 months. Patients were medically treated upon admission, based on the recommendation of their physician. Patients suffering from psychotic mood disorders were excluded from the study.

Healthy controls over 18 years of age were recruited through advertisements unless: They had a personal history of any psychiatric disorder or had a positive family history of psychotic disorders in first- or second-degree relatives.

If a subject in either group (patient or control) had a history of neurological or cerebrovascular disorders or any MRI contraindications, they were excluded from the study.

The study was carried out in accordance with the latest version of the Declaration of Helsinki. The study design was reviewed and approved by the Research Ethics Board. Each participant received a complete description of the study and provided written informed consent.

Data were acquired at the National Centre of Mental Health in Klecany, Czech Republic. The data were acquired at the National Institute of Mental Health using Siemens MAGNETOM Prisma 3 T. The acquisition parameters of T1-weighted images using MPRAGE sequence were: 240 scans; slice thickness: 0.7 mm; repetition time: 2400 ms; echo time: 2.34 ms; inversion time: 1000 ms; flip angle: 8°; and acquisition matrix: 320 mm × 320 mm.

## Preprocessing and analysis

Prior to normative modelling, all T1 images were preprocessed using the FreeSurfer v.(7.2) recon-all pipeline. While in the context of longitudinal analysis the longitudinal FreeSurfer preprocessing pipeline is appropriate, we additionally performed cross-sectional preprocessing (*Reuter et al., 2012*). The reason to conduct this analysis is threefold: First, the impact of preprocessing on the *z*-scores of normative models lacks prior investigation. Second, the training dataset of 58,000 subjects initially underwent cross-sectional preprocessing, introducing a methodological incongruity. Third, certain large-scale studies, constrained by computational resources, exclusively employ cross-sectional preprocessing. Understanding the consistency of results between the two approaches becomes crucial in such cases.

In line with *Rutherford et al., 2022*, we performed a simple quality control procedure whereby all subjects having a rescaled Euler number greater than 10 were labelled outliers and were not included in the analysis (*Table 1*) (see *Rutherford et al., 2022*, and *Kia et al., 2022*, for further details).

After preprocessing, patient data were projected into the adapted normative model (median Rho across all IDP was 0.3 and 0.26 for the first and the second visit, respectively—see *Appendix 4— figure 1*). The pre-trained model used for adaptation was the lifespan_58_82_sites (*Rutherford et al., 2022*). For each subject and visit, we obtained cross-sectional $z$-score, as well as the underlying values needed for its computation, particularly $\varphi(y)$ and $\bar{\mathbf{w}}^T \phi(\mathbf{x})$. We conducted a cross-sectional analysis of the original $z$-scores to evaluate each measurement independently. We then tested for the difference of the cross-sectional $z$-scores $z^{(1)} - z^{(2)}$ between the patients and held-out controls using Mann-Whitney U test and corrected for multiple tests using the Benjamini-Hochberg FDR correction at the 5% level of significance.

Subsequently, following *Equation 11*, we derived the *z-diff* scores of change between visits. We conducted two analyses: one to investigate the group-level effect and another to link the *z-diff* to the longitudinal changes in clinical scales.

At a group level, we identified regions with *z-diff* scores significantly different from zero using the Wilcoxon test, accounting for multiple comparisons using the Benjamini-Hochberg FDR correction.

Additionally, we performed a more traditional longitudinal analysis. As all visits were approximately 1 year apart, we conducted an analysis of covariance (ANCOVA). The ANCOVA model combines a general linear model and ANOVA. Its purpose is to examine whether the means of a dependent variable (thickness in visit 2) are consistent across levels of a categorical independent variable (patients or controls) while accounting for the influences of other variables (age, gender, and thickness in visit 1). We conducted a separate test for each IDP and controlled the relevant p-values across tests using the FDR correction.

For linking the *z-diff* score to clinical longitudinal change, we transformed the *z-diff* score across all IDPs using PCA to decrease the dimensionality of the data as well as to avoid fishing. We ran PCA with 10 components and using Spearman correlation related the scores with changes in the Positive and Negative Syndrome Scale (PANSS) and Global Assessment of Functioning (GAF) scale.

## Results

### Effect of preprocessing

After obtaining cross-sectional $z$-scores for both types of preprocessing, we visually observed a decrease in variance between the two visits in longitudinal preprocessing compared to the cross-sectional one (*Figure 4*). More specifically, we calculated the mean of the difference between $z$-scores of visit 2 and visit 1 for each individual IDP, stratified by preprocessing and group, across all subjects. We then visualised the distribution of these means using a histogram (*Figure 4C*). Alternatively, we also computed the mean difference between $z$-scores of visit 2 and visit 1 across all IDPs for each subject, and plotted a histogram of these values. Note that this step was only done to estimate the effect of preprocessing on $z$-scores for further discussion. Its impact on the results is elaborated on in the Discussion.

### Cross-sectional results

At a group level, patients had significantly lower thicknesses in most areas compared to healthy populations. In particular, this difference was distinct even in the first visit, indicating structural changes prior to diagnosis (*Figure 5*).

### Longitudinal results and patterns of change

A longitudinal analysis that evaluated the amount of structural change between the two visits showed a significant cortex normalisation of several frontal areas, namely the right and left superior frontal sulcus, the right and left middle frontal sulcus, the right and left middle frontal gyrus, and the right superior frontal gyrus (*Figure 6*).

In terms of linking longitudinal change in clinical scores with changes captured by *z-diff* scores, each of the two scales was well correlated with different component. The first PCA component, which itself reflected the average change in global thickness across patients, was correlated with the change

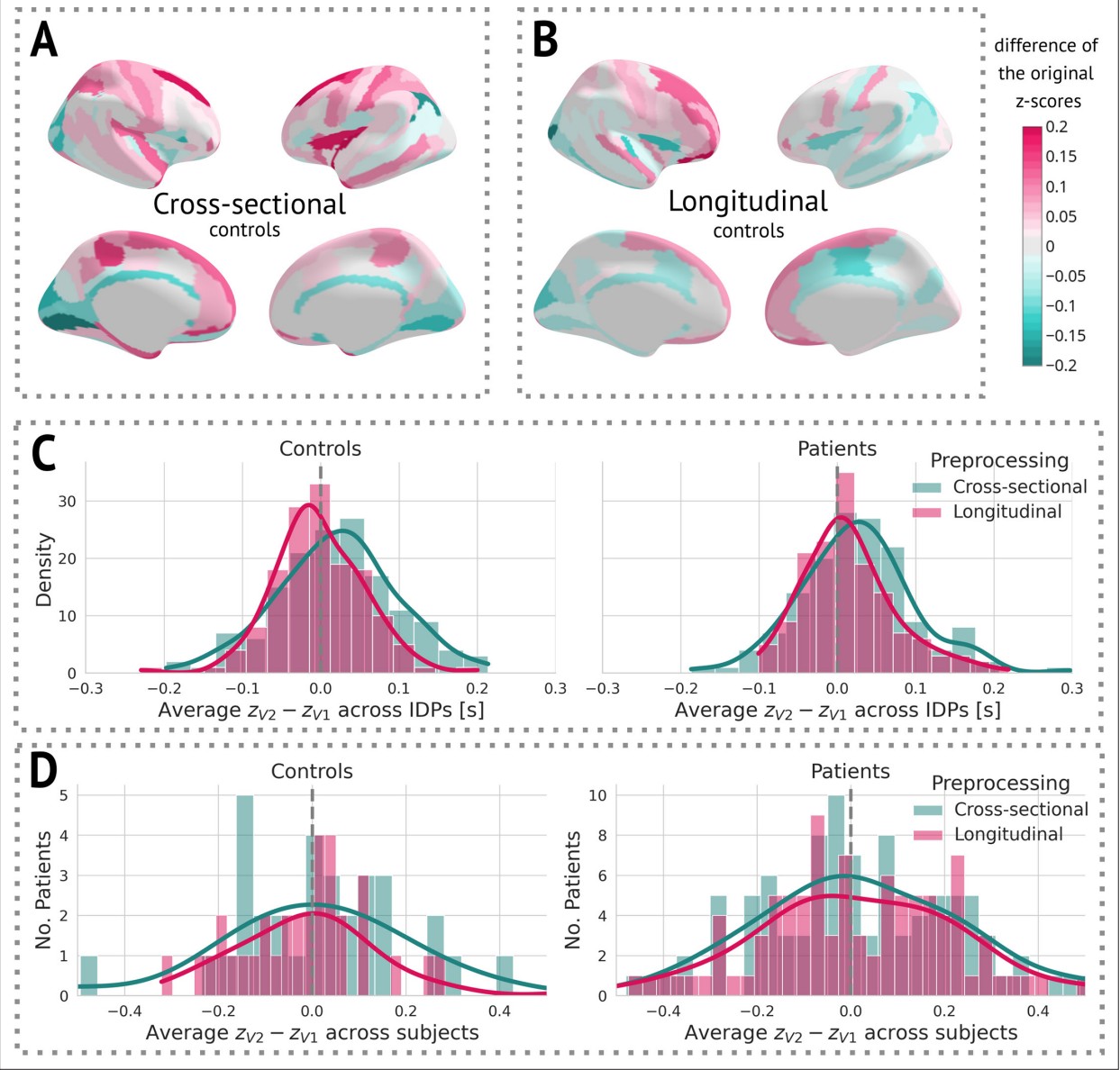

**Figure 4.** The effect of preprocessing across all subjects and image-derived phenotypes (IDPs). (**A**) Cross-sectional preprocessing: Heatmap of the difference of the original $z$-scores ($z^{(2)} - z^{(1)}$) on held-out controls. (**B**) Longitudinal preprocessing: Heatmap of the difference of the original $z$-scores ($z^{(2)} - z^{(1)}$) on held-out controls. (**C**) Histogram of the average ($z$) across all IDPs stratified by health status and preprocessing. (**D**) Histogram of the average ($z^{(2)} - z^{(1)}$) of each subject stratified by health status and preprocessing.

in GAF score, whereas the second component significantly correlated with the change in PANSS score (see *Figure 7*).

## Discussion

Longitudinal neuroimaging studies allow us to assess the effectiveness of interventions and gain deeper insights into the fundamental mechanisms of underlying diseases. Despite the significant expansion of our knowledge regarding population variation through the availability of publicly accessible neuroimaging data, this knowledge, predominantly derived from cross-sectional observations, has not been adequately integrated into methods for evaluating longitudinal changes.

We propose an analytical framework that builds on normative modelling and generates unbiased features that quantify the degree of change between visits, whilst capitalising on information extracted from large cross-sectional cohorts.

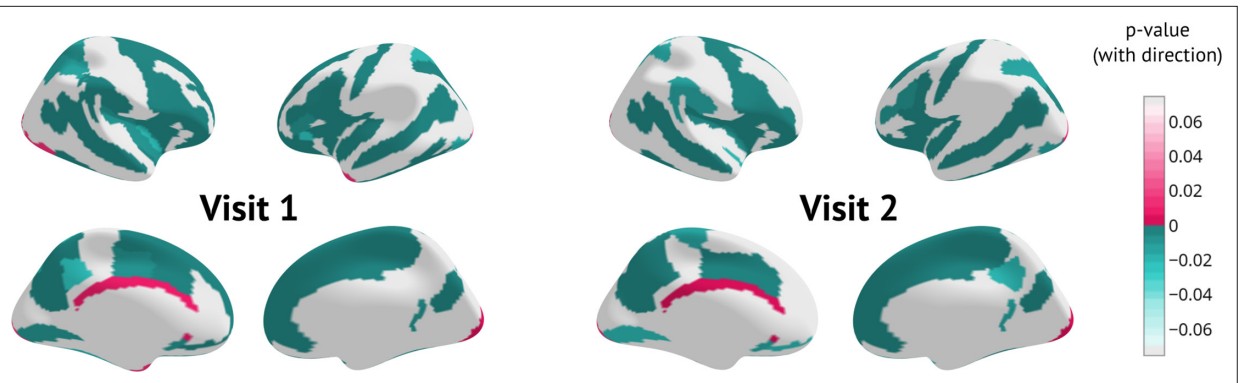

**Figure 5.** Cross-sectional results for each visit separately: p-Values of Mann-Whitney U test between patients and held-out controls surviving Benjamini-Hochberg correction. The sign indicates the direction of change (negative means lower thickness in patients).

## Methodological contribution

Our approach is rooted in the normative modelling method based on Bayesian regression (*Fraza et al., 2021*), the pre-trained version of which recently became available (*Rutherford et al., 2022*). We showed that the estimation of longitudinal changes is available based on a preexisting cross-sectional normative model and only requires a set of healthy controls on which the variance of healthy change might be estimated. We denoted the score obtained after running the procedure as a *z-diff* score, which quantifies the extent of change between visits beyond what one would expect in the healthy population.

To this end, our approach implies that in a group of healthy controls, we should observe only change that is consistent with the healthy population, i.e., zero average *z-diff* score. We used the data of 33 healthy controls which were originally used for the site-specific adaptation (for more details, see the discussion part on implementation) and computed their *z-diff* scores. After averaging these scores across all subjects, the *z-diff* score of no region was statistically significant from zero (after FDR correction). However, as pointed out by a recent work (*Di Biase et al., 2023*) studying the effect of cross-sectional normative models on longitudinal predictions, the cross-sectionally derived population centiles *by design* lack information about longitudinal dynamics. Consequently, what may appear as a population-level trajectory does not necessarily align with individual subjects' actual trajectories. Although it is important to keep this caveat in mind, it can be fully addressed only by proper longitudinal normative models, which is beyond the scope of this paper.

Instead, we argue that the population-level trajectory carries meaningful information about individual-level trajectories, and we allow for a flexible process of deviations between the two. By estimating the amplitude of the longitudinal change in healthy controls (adjusting for the population-level trajectory), we get an insight into this process. Naturally, if the healthy changes have a high amplitude (corresponding to low to negative $\rho$ in the 'Simulation' section), it becomes more challenging to identify subjects who actually diverge from the 'healthy' trajectory, i.e., the *z-diff* score becomes overly conservative. A potential reason for the high-amplitude residual process is substantial acquisition or processing noise. As evident from the clinical findings, only a fraction of subjects were identified as having undergone significant changes (*Appendix 5—figure 1*). However, at the group level, the significance of the observed changes persisted. Therefore, while the method adopts a cautious approach when assessing individual changes, it identifies effectively group-level changes. Note that this is not unique to our method, but is rather a general statistical feature.

Furthermore, unlike in *Di Biase et al., 2023*, our approach does not aim to predict individual trajectories, but rather to quantify whether the observed changes over time exceed what would be expected.

## Implementation

At the implementation level, our approach requires two stages of adaptation: site-specific adaptation, as presented in *Rutherford et al., 2022*, and a second level where we compute the variance of healthy longitudinal change (noise) in healthy controls. However, if the number of longitudinal controls

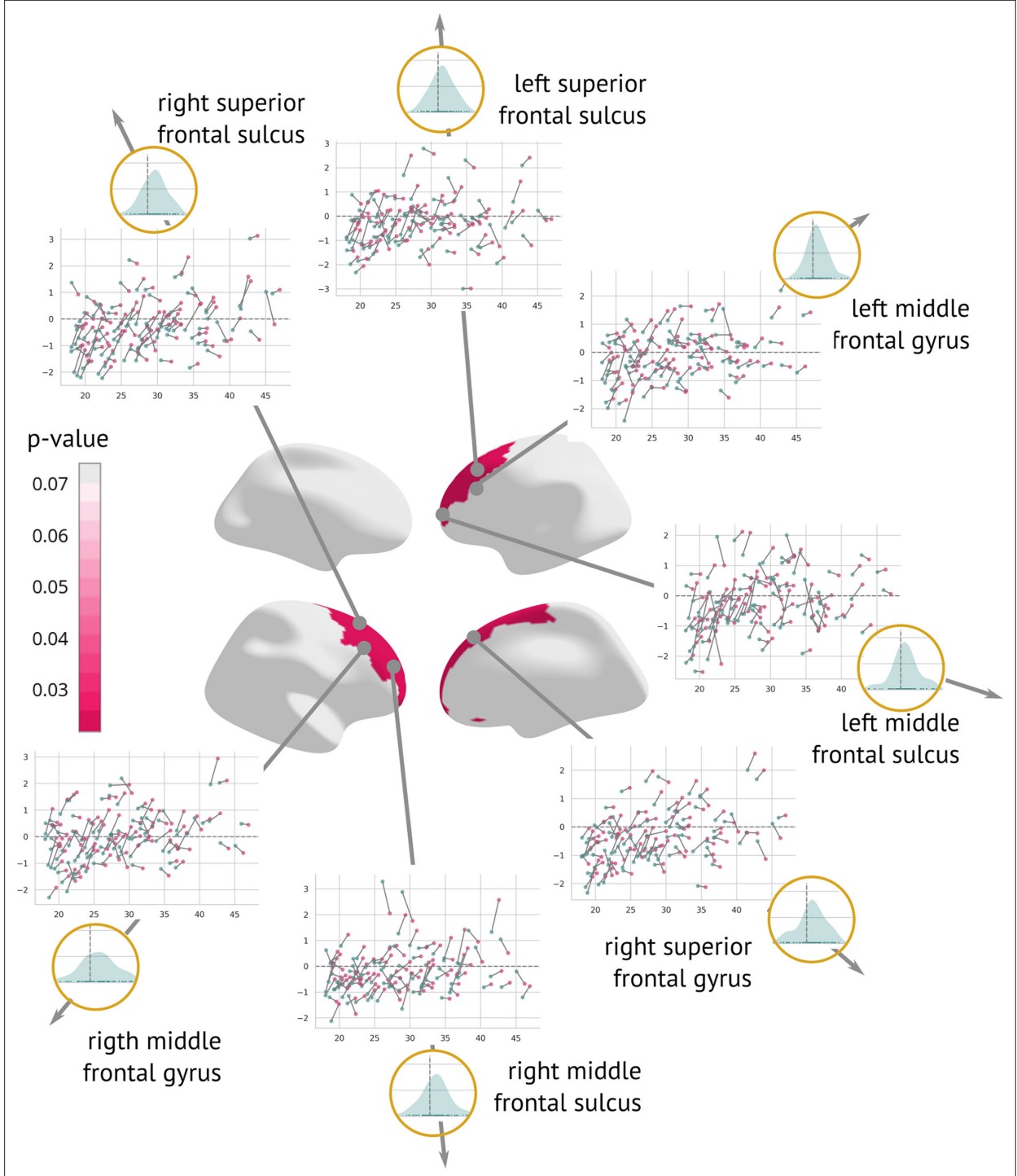

**Figure 6.** Regions significantly changed between the visits: Map of regions significantly changed between the two visits (centre). Each region is described using a scatter plot of $z$-scores across all patients for both visits (the $x$-axis describes age and the $y$-axis depicts the $z$-score. Blue dots represent the first and pink dots represent the second visit). The grey dashed line highlights $z = 0$. Histograms in the golden circles depict the distribution of the $z$-diff score.

is limited, the site-specific adaptation may be omitted. The purpose of site-specific adaptation is to generate unbiased cross-sectional $z$-scores that are zero-centred with a variance of one for healthy controls. However, in the case of longitudinal analysis, the offset and normalisation constant are irrelevant since they will be identical for both visits. Therefore, the estimation of healthy longitudinal

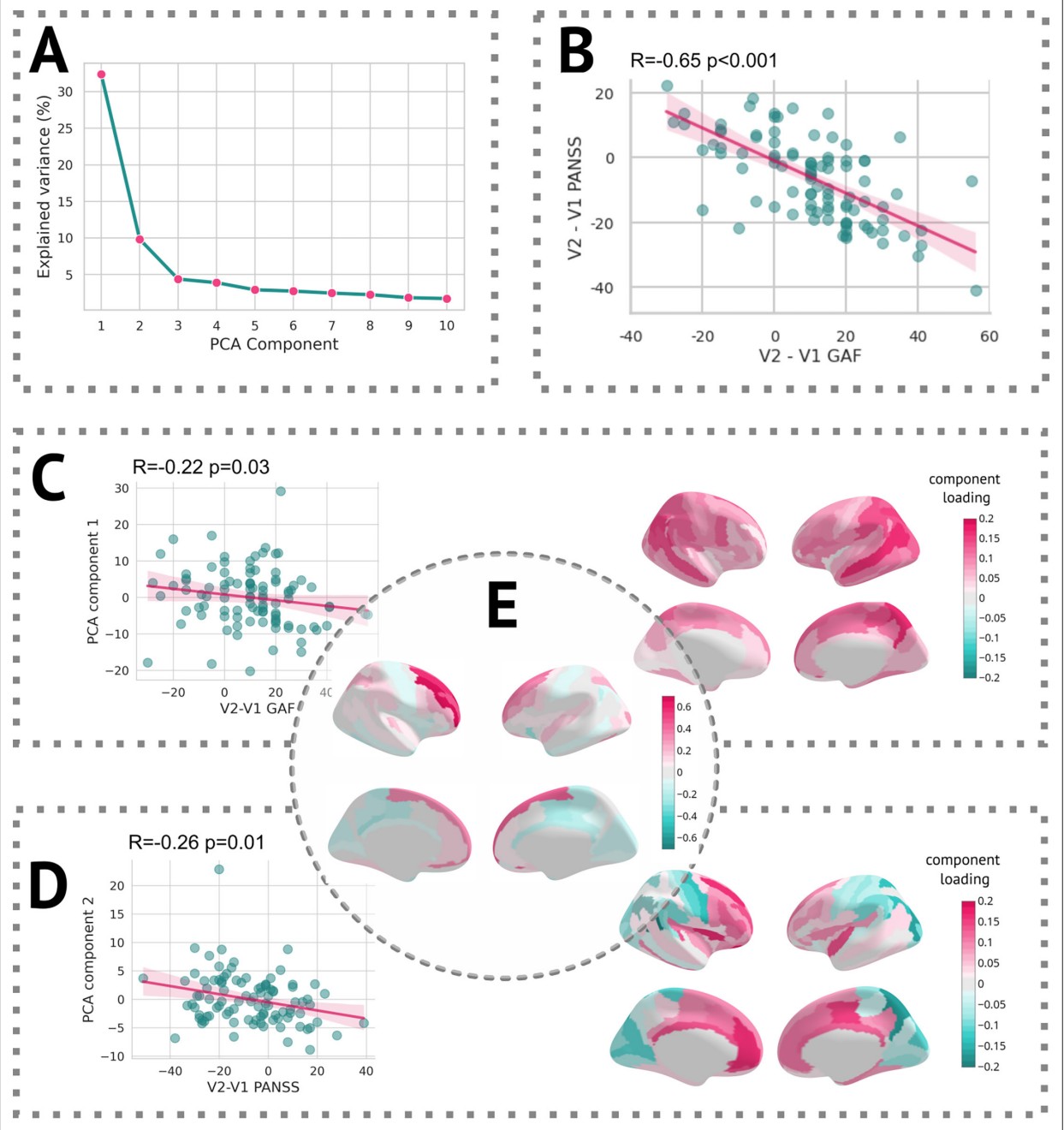

**Figure 7.** Results of the PCA. (**A**) Scree plot of the explained variance of PCA components. (**B**) Scatter plot of change in the Global Assessment of Functioning (GAF) scale vs. the change in the Positive and Negative Syndrome Scale (PANSS) scale. (**C**, left) Scatter plot of the first PCA component and difference in the GAF scale. (**C**, right) Heatmap of PCA loadings for the first component. (**D**, left) Scatter plot of the second PCA component and difference in the PANSS scale. (**D**, right) A heatmap of PCA loadings for the second component. (**E**) Average *z-diff* score.

change is the only essential factor in producing the *z-diff* score. Note that in this scenario, the cross-sectional result should not be interpreted.

## Clinical results

Examination of the effect of preprocessing on *z*-scores showed that longitudinal preprocessing indeed decreases intra-subject variability compared to cross-sectional preprocessing. However, to assess the added benefit of the preprocessing, we also computed the core results (regions that significantly changed in time) for the cross-sectional data. The significant results were mostly consistent with a

longitudinal pipeline: Six out of seven originally significant regions were still statistically significant (with the exception of the right middle frontal sulcus), and three other regions were labelled significant: the left superior frontal gyrus, the right inferior frontal sulcus, and the right medial or olfactory orbital sulcus (*Appendix 5—figure 2*). Therefore, it is also possible to use cross-sectional preprocessing for longitudinal analysis, however, at a cost of increased between-visit variance and consequently decreased power (in comparison to the longitudinal preprocessing).

The observation of cortical normalisation between the visits of early schizophrenia patients is, to a degree, counterintuitive to the historical narrative, which mostly assumes grey matter thinning. There is now increasing evidence that: (i) trajectories of cortical thickness are highly variable across different individuals after the first psychotic episode and (ii) that individuals treated with second-generation antipsychotics and with careful clinical follow-up can show normalisation of cortical thickness atypicalities after the first episode (*Berthet et al., 2024*; *Canal-Rivero et al., 2023*). In *Berthet et al., 2024*, a cohort of 79 first-episode psychosis patients were longitudinally monitored with two follow-ups, after a year and 10 years. Although cross-sectionally, patients showed significantly lower (cross-sectional) z-scores at baseline (which is consistent with our findings), their proportion decreased over time, indicating an attenuation of differences over time. Canal et al. reported similar observation in larger cohort (*Canal-Rivero et al., 2023*) of 357 people with first-episode psychosis followed over 10-year period. Notably, no changes in cortical thickness were observed within the first 3 years. Afterwards, the trajectories started diverging, with cortical thinning observed only in people who experienced worsening of negative symptoms on the expressivity dimension of Scale for the Assessment of Negative Symptoms.

Furthermore, a meta-analysis of 50 longitudinal studies examining individuals with a heightened risk of psychosis revealed that 15 of the 19 studies indicated deviations in grey matter developmental trajectories between those with persistent symptoms and those whose symptoms resolved (*Merritt et al., 2021*). The authors propose that grey matter developmental trajectories may return to normal levels in individuals in the High-Risk Remitting group by early adulthood, whereas neurological irregularities may continue to advance in those whose symptoms do not resolve. Although our cohort had already received a diagnosis of schizophrenia, it is possible that early identification and treatment supported these compensatory mechanisms, as demonstrated by the normalisation of grey matter thickness in frontal regions. Notably, the affected regions also increased in raw grey matter thickness (as measured in mm, see *Appendix 5—figure 3*).

Additionally, we observed significant correlations between the PCA components of the *z-diff* score and longitudinal changes in clinical scales, as illustrated in *Figure 7*. Notably, each clinical scale exhibited distinct associations with separate PCA components, despite substantial intercorrelations (*Figure 7B*).

The first PCA component, which predominantly captured global changes in grey matter thickness, displayed a negative correlation with improvements in the GAF score (*Figure 7C*). This unexpected inverse relationship would suggest that patients who demonstrated clinical improvement over time exhibited a more pronounced decrease in grey matter thickness, as quantified by the *z-diff* score. However, further investigation revealed that this correlation was primarily driven by the patients' GAF scores in the initial visit. Specifically, the correlation between GAF scores at the first visit and the first PCA component yielded a coefficient of $R = 0.19$ ($p = 0.06$), whereas the correlation with scores at the second visit was $R = -0.10$ ($p = 0.31$). These findings suggest that lower GAF scores during the initial visit are predictive of subsequent grey matter thinning.

Conversely, the interpretation of the second PCA component, significantly correlated with changes in the PANSS score, was more straightforward (*Figure 7D*). The observed normalisation of grey matter thickness in frontal areas was positively correlated with improvements in the PANSS scale, indicating that symptom amelioration was accompanied by the normalisation of grey matter thickness in these regions.

Finally, we conducted an analysis of longitudinal change using conventional statistical approaches to compare the results with normative modelling. Out of 148 areas tested by ANCOVA, 6 were statistically significant. However, after controlling for multiple comparisons, no IDP persisted. This result highlights the advantages of normative models and shows improved sensitivity of our method in comparison with more conventional approaches.

## Limitations

Estimating the intra-subject variability is a complex task that might be affected by acquisition and physiological noise. Assumptions must be made about the longitudinal behaviour of healthy subjects. The former problem is unavoidable, whereas the latter might be addressed by constructing longitudinal normative models. However, the project necessary for such a task would have to map individuals across their lifespan consistently. The efforts to create such a dataset are already in progress through projects like the ABCD study (*Casey et al., 2018*), but much more data are still needed to construct a full-lifespan longitudinal model.

Additionally, the *z-diff* score only quantifies the size of the change irrespective of the initial position (e.g. cross-sectional z-score being above or below 0). However, in subsequent analyses, it is possible to construct models that include both, the original (cross-sectional) position combined with the (longitudinal) change. Indeed, the non-random sampling of large cohort studies is a challenge for nearly all studies using such cohorts, regardless of the statistical approach used.

Finally, our clinical results may be affected by selection bias, where subjects experiencing a worsening of their condition dropped out of the study, whereas patients with lower genetic risk or more effective treatment continued to participate.

## Conclusion

We have developed a method that utilises pre-trained normative models to detect unusual longitudinal changes in neuroimaging data. Our approach offers a user-friendly implementation and has demonstrated its effectiveness through a comprehensive analysis. Specifically, we observed significant grey matter changes in the frontal lobe of schizophrenia patients over time, surpassing the sensitivity of conventional statistical approaches. This research represents a significant advancement in longitudinal neuroimaging analysis and holds great potential for further discoveries in neurodegenerative disorders.

## Acknowledgements

This research was supported by the Czech Health Research Council (NU21-08-00432); Programme Johannes Amos Comenius ('BRADY' CZ.02.01.01/00/22_008/0004643); European Research Council (grant 'MENTALPRECISION', 10100118), the Wellcome Trust under an Innovator awards ('BRAINCHART', 215698/Z/19/Z and 'PRECOGNITION', 226706/Z/22/Z), the Ministry of Education, Youth and Sports (CZ.02.2.69/0.0/0.0/18_053/0017594); and the Czech Technical University Internal Grant Agency (SGS22/062/OHK3/1 T/13).

## Additional information

### Competing interests

Andre F Marquand: Reviewing editor, eLife. The other authors declare that no competing interests exist.

### Funding

| Funder | Grant reference number | Author |
| --- | --- | --- |
| Czech Health Research Council | NU21-08-00432 | Filip Španiel<br>Jaroslav Hlinka |
| Programme Johannes Amos Comenius | 'BRADY' CZ.02.01.01/00/22_008/0004643 | Jaroslav Hlinka |
| European Research Council | 'MENTALPRECISION': 10100118 | Andre F Marquand |
| Wellcome Trust | 'BRAINCHART': 215698/Z/19/Z | Andre F Marquand |
| Wellcome Trust | 'PRECOGNITION': 226706/Z/22/Z | Andre F Marquand |

| Funder | Grant reference number | Author |
|---|---|---|
| Ministry of Education, Youth and Sports | CZ.02.2.69/0.0/0.0/18_053/00 17594 | Barbora Rehak Buckova |
| Czech Technical University Internal Grant Agency | SGS22/062/OHK3/1 T/13 | Barbora Rehak Buckova |

The funders had no role in study design, data collection and interpretation, or the decision to submit the work for publication. For the purpose of Open Access, the authors have applied a CC BY public copyright license to any Author Accepted Manuscript version arising from this submission.

## Author contributions

Barbora Rehak Buckova, Conceptualization, Data curation, Software, Formal analysis, Funding acquisition, Validation, Investigation, Visualization, Methodology, Writing – original draft, Project administration, Writing – review and editing; Charlotte Fraza, Software, Investigation, Methodology, Writing – original draft, Writing – review and editing; Rastislav Rehák, Formal analysis, Investigation, Methodology, Writing – original draft, Writing – review and editing; Marián Kolenič, Data curation, Investigation, Writing – original draft, Writing – review and editing; Christian F Beckmann, Resources, Supervision, Writing – review and editing; Filip Španiel, Resources, Data curation, Funding acquisition, Project administration, Writing – review and editing; Andre F Marquand, Conceptualization, Resources, Supervision, Funding acquisition, Investigation, Methodology, Writing – original draft, Project administration, Writing – review and editing; Jaroslav Hlinka, Conceptualization, Resources, Formal analysis, Supervision, Funding acquisition, Investigation, Methodology, Writing – original draft, Project administration, Writing – review and editing

## Author ORCIDs

Barbora Rehak Buckova (ID) https://orcid.org/0000-0001-5619-3946
Charlotte Fraza (ID) https://orcid.org/0000-0002-7088-9250
Marián Kolenič (ID) https://orcid.org/0000-0002-2382-3478
Andre F Marquand (ID) https://orcid.org/0000-0001-5903-203X
Jaroslav Hlinka (ID) https://orcid.org/0000-0003-1402-1470

## Ethics

This study is based on data from the Early-Stage Schizophrenia Outcome (ESO) study, a prospective trial conducted in the Prague and Central Bohemia surveillance area investigating first-episode schizophrenia spectrum subjects. The study was conducted in compliance with the Declaration of Helsinki and was approved by the Research Ethics Board of NIMH Klecany, approval No. 127/17. All participants received detailed information about the purpose of the study, the experimental procedures, and their right to withdraw at any time without consequence. Written informed consent was obtained from all participants prior to their participation.

Reviewer #2 (Public review): https://doi.org/10.7554/eLife.95823.4.sa1
Author response https://doi.org/10.7554/eLife.95823.4.sa2

# Additional files

## Supplementary files

MDAR checklist

## Data availability

The raw data cannot be shared publicly, as this study involves human participants whose consent did not include public sharing of raw neuroimaging data. However, requests for anonymized raw data can be directed to Filip Španiel (filip.spaniel@nudz.cz) and will be reviewed by an independent data access committee. Requests must include a research proposal, intended data use, and agreement to a data transfer agreement to ensure participant confidentiality. All procedures comply with EU General Data Protection Regulation (GDPR) and local ethical standards. The processed version of the dataset,

i.e. higher-level longitudinal thickness features extracted using FreeSurfer are also publicly available (https://github.com/predictive-clinical-neuroscience/PCNtoolkit-demo/tree/main/tutorials/Long_NM), accompanied by a full tutorial for replicating the analysis. In particular, specific code/software used to analyse the data is publicly available at https://github.com/predictive-clinical-neuroscience/PCNtoolkit-demo/tree/main/tutorials/Long_NM (*Rutherford et al., 2024*).

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

## Appendix 1

### Posterior predictive distribution for difference between visits

Here, we derive the posterior predictive distribution for the difference $\varphi(y^{(2)}) - \varphi(y^{(1)})$. The argument is standard. Denote $\Delta_{\mathbf{x}} = \phi(\mathbf{x}^{(2)}) - \phi(\mathbf{x}^{(1)})$ and $\Delta_y = \varphi(y^{(2)}) - \varphi(y^{(1)})$. Since $\Delta_{\mathbf{x}}^T \mathbf{w} | \mathbf{x}^{(1)}, \mathbf{x}^{(2)}; \mathbf{y}, \mathbf{\Phi}; \omega^2, \sigma^2, \gamma \sim \mathcal{N}(\Delta_{\mathbf{x}}^T \bar{\mathbf{w}}, \Delta_{\mathbf{x}}^T \mathbf{A}^{-1} \Delta_{\mathbf{x}})$ and $\Delta_y | \mathbf{x}^{(1)}, \mathbf{x}^{(2)}; \mathbf{w} \sim \mathcal{N}(\Delta_{\mathbf{x}}^T \mathbf{w}, 2\sigma_\xi^2)$, the posterior predictive density is

$$f(\Delta_y | \mathbf{x}^{(1)}, \mathbf{x}^{(2)}; \mathbf{y}, \mathbf{\Phi}; \omega^2, \sigma^2, \gamma) =$$

$$= \int f_{\mathcal{N}(\Delta_{\mathbf{x}}^T \mathbf{w}, 2\sigma_\xi^2)}(\Delta_y | \mathbf{x}^{(1)}, \mathbf{x}^{(2)}; \mathbf{w}) \cdot f_{\mathcal{N}(\Delta_{\mathbf{x}}^T \bar{\mathbf{w}}, \Delta_{\mathbf{x}}^T \mathbf{A}^{-1} \Delta_{\mathbf{x}})}(\Delta_{\mathbf{x}}^T \mathbf{w} | \mathbf{x}^{(1)}, \mathbf{x}^{(2)}; \mathbf{y}, \mathbf{\Phi}; \omega^2, \sigma^2, \gamma)\, d(\Delta_{\mathbf{x}}^T \mathbf{w})$$

$$= \int f_{\mathcal{N}(0, 2\sigma_\xi^2)}(\Delta_y - \Delta_{\mathbf{x}}^T \mathbf{w} | \mathbf{x}^{(1)}, \mathbf{x}^{(2)}; \mathbf{w}) \cdot f_{\mathcal{N}(\Delta_{\mathbf{x}}^T \bar{\mathbf{w}}, \Delta_{\mathbf{x}}^T \mathbf{A}^{-1} \Delta_{\mathbf{x}})}(\Delta_{\mathbf{x}}^T \mathbf{w} | \mathbf{x}^{(1)}, \mathbf{x}^{(2)}; \mathbf{y}, \mathbf{\Phi}; \omega^2, \sigma^2, \gamma)\, d(\Delta_{\mathbf{x}}^T \mathbf{w}).$$

This has the familiar convolution form of the densities of $\mathcal{N}(0, 2\sigma_\xi^2)$ and $\mathcal{N}(\Delta_{\mathbf{x}}^T \bar{\mathbf{w}}, \Delta_{\mathbf{x}}^T \mathbf{A}^{-1} \Delta_{\mathbf{x}})$. It is known to produce the density of $\mathcal{N}(\Delta_{\mathbf{x}}^T \bar{\mathbf{w}}, \Delta_{\mathbf{x}}^T \mathbf{A}^{-1} \Delta_{\mathbf{x}} + 2\sigma_\xi^2)$ (by completion to squares in the exponent).

# Appendix 2

## Estimates of autocorrelation from the data

The simulation study demonstrated that the most pronounced advantages of the *z-diff* score over the *z*-score subtraction occur when the degree of autocorrelation $\rho$ is above 0.5. To determine whether this scenario reflects real-world data, we derive what values of autocorrelation are implied by our data. Specifically, in the pool of our controls from our local dataset, which is further used to illustrate the use of the method, we combine the estimate $\widehat{2\sigma^2(1-\rho)}$ from *Equation 14* with the cross-sectional estimate $\widehat{\sigma}^2$ (from the first visit) to compute

$$\hat{\rho} = 1 - \frac{\widehat{2\sigma^2(1-\rho)}}{2\widehat{\sigma}^2} \tag{19}$$

*Appendix 2—figure 1* presents a histogram of $\hat{\rho}$ values across all IDPs in our dataset. The results clearly show that real-world data exhibit a very high degree of autocorrelation, further strengthening the justification for using *z-diff* scores.

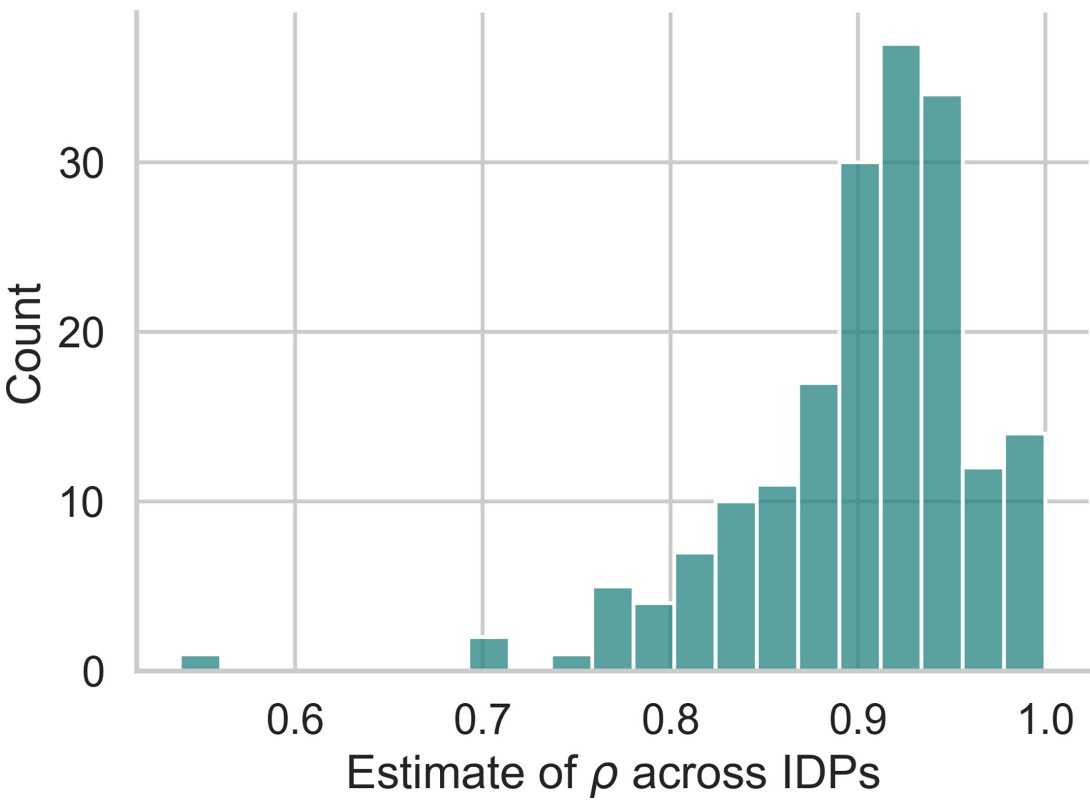

**Appendix 2—figure 1.** $\hat{\rho}$ estimates derived from the data.

## Appendix 3

### Comparison of *z-diff* and *z*-score subtraction across different variances

As described by the generalised dynamics (**Equation 18**), three factors influence the detection rate: the magnitude of the disruption ($\Delta$), the level of autocorrelation ($\rho$), and the variance in population ($\sigma^2$). While the main article focuses on the first two factors, here, we examine how the variance impacts the detection difference between the *z-diff* score and the subtraction of individual *z*-scores.

The simulation results in **Appendix _3—figure 1** confirm that the general conclusions from the main article remain valid. Notably, the superiority of the *z-diff* score becomes even more pronounced as the variance increases.

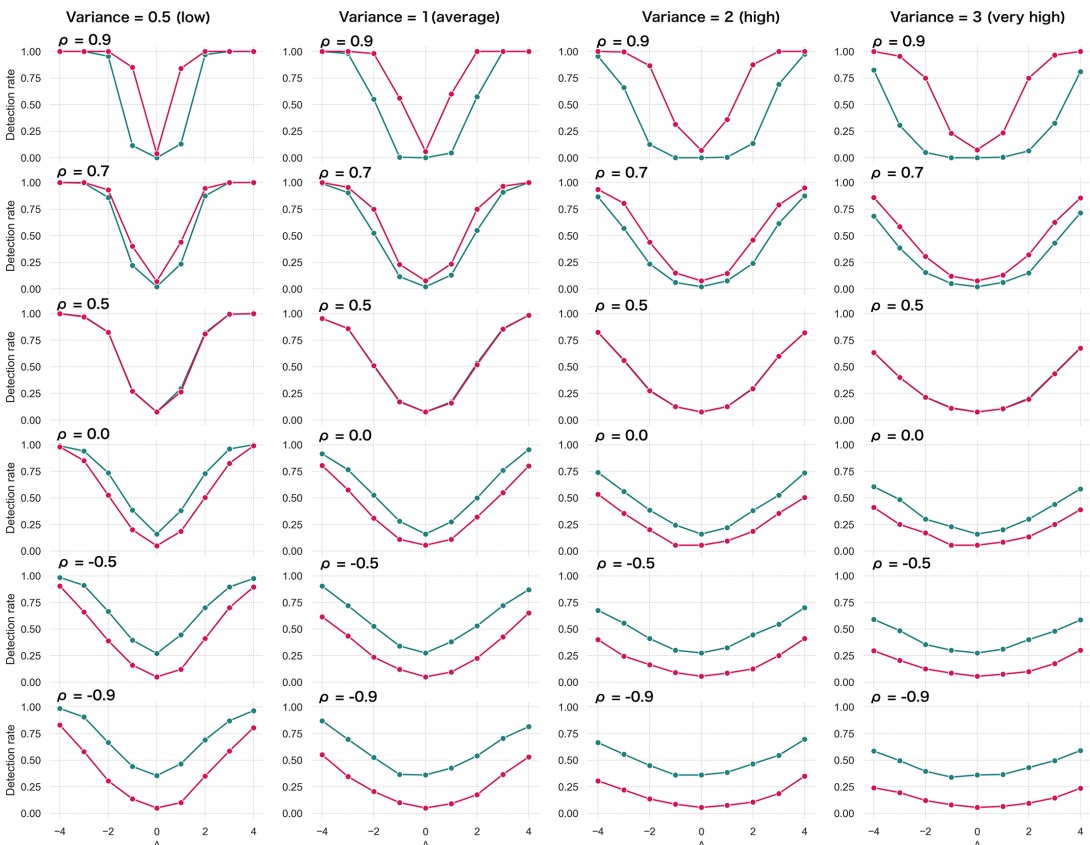

**Appendix 3—figure 1.** Probability of detecting a true disruption $\Delta$ for various values of autocorrelation $\rho$ (rows) and variances $\sigma^2$ (columns) comparing the performance of our *z-diff* method against the naïve subtraction of *z*-scores. We use $\theta = 0.05$.

## Appendix 4

## Quality of fit across regions of interest

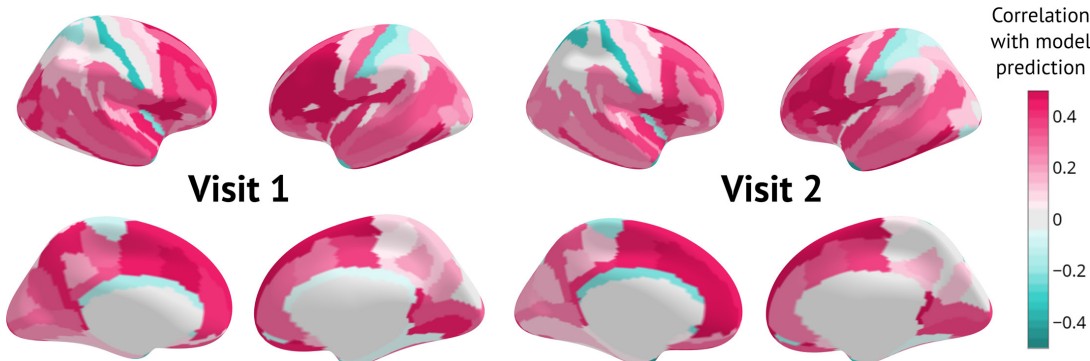

**Appendix 4—figure 1.** Quality of fit as measured by Rho for the first and the second visit.

## Appendix 5

### Comparison of preprocessing

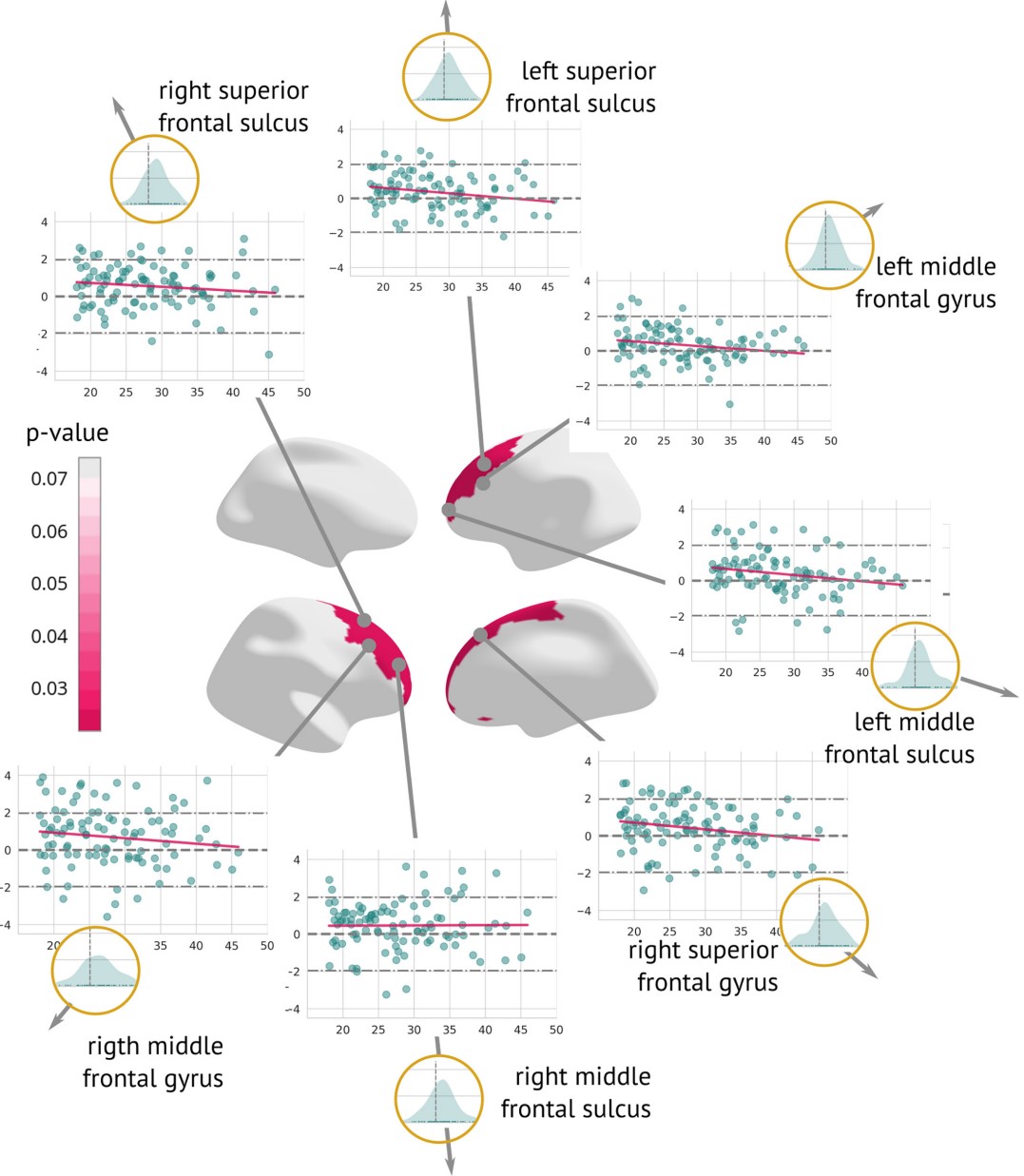

**Appendix 5—figure 1.** Regions significantly changed between the visits (longitudinal preprocessing): Map of regions significantly changed between the two visits (centre). Each region is described using a scatter plot of *z-diff* across all patients for both visits (the *x*-axis describes age and the *y*-axis depicts the *z-diff*. Blue dots represent individual patients and the pink line shows a trend of *z-diff* change). The grey dashed line highlights *z*=0. Histograms in the golden circles depict the distribution of the *z-diff* score.

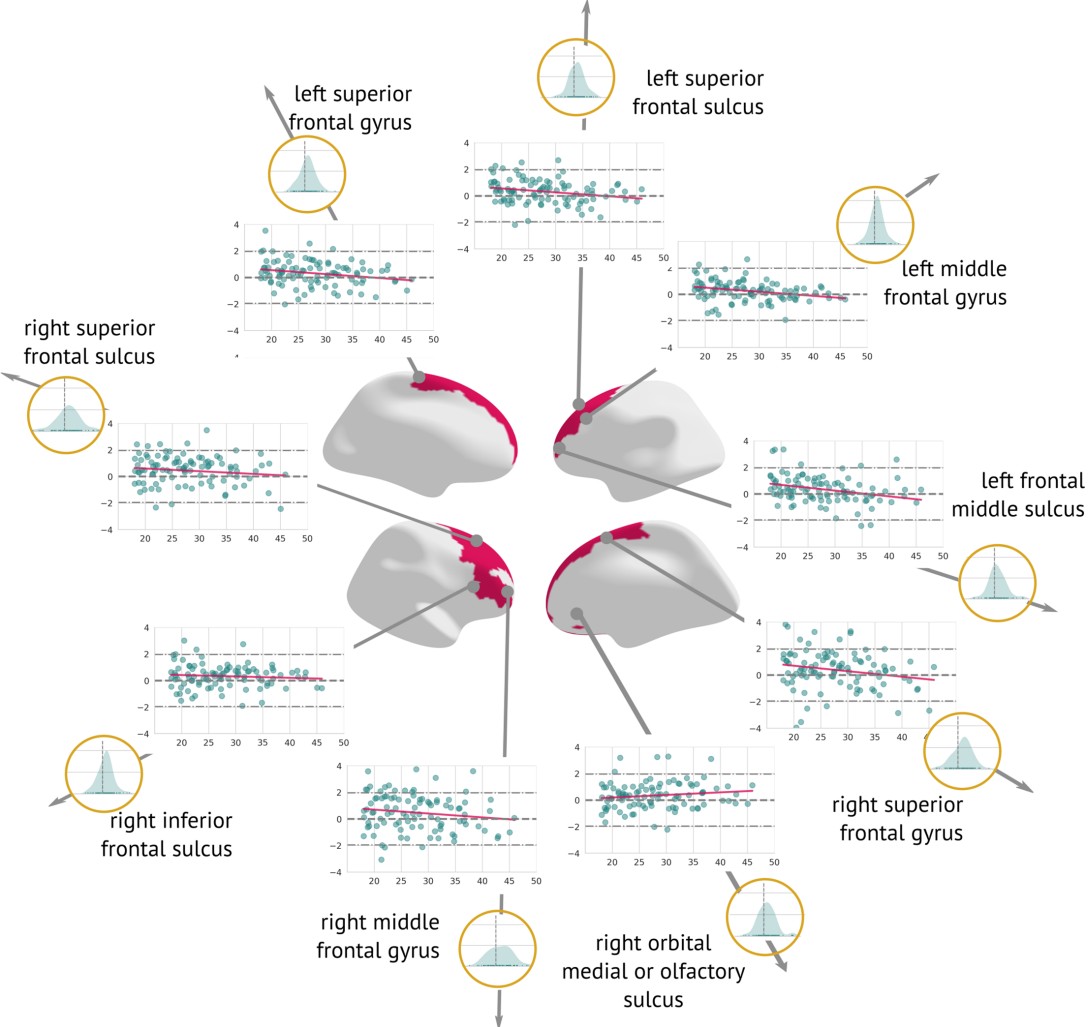

**Appendix 5—figure 2.** Regions significantly changed between the visits (cross-sectional preprocessing): Map of regions significantly changed between the two visits (centre). Each region is described using a scatter plot of *z-diff* scores across all patients for both visits (the *x*-axis describes age and the *y*-axis depicts the *z-diff* score). The grey dashed line highlights $z=0$. Histograms in the golden circles depict the distribution of the *z-diff* score.

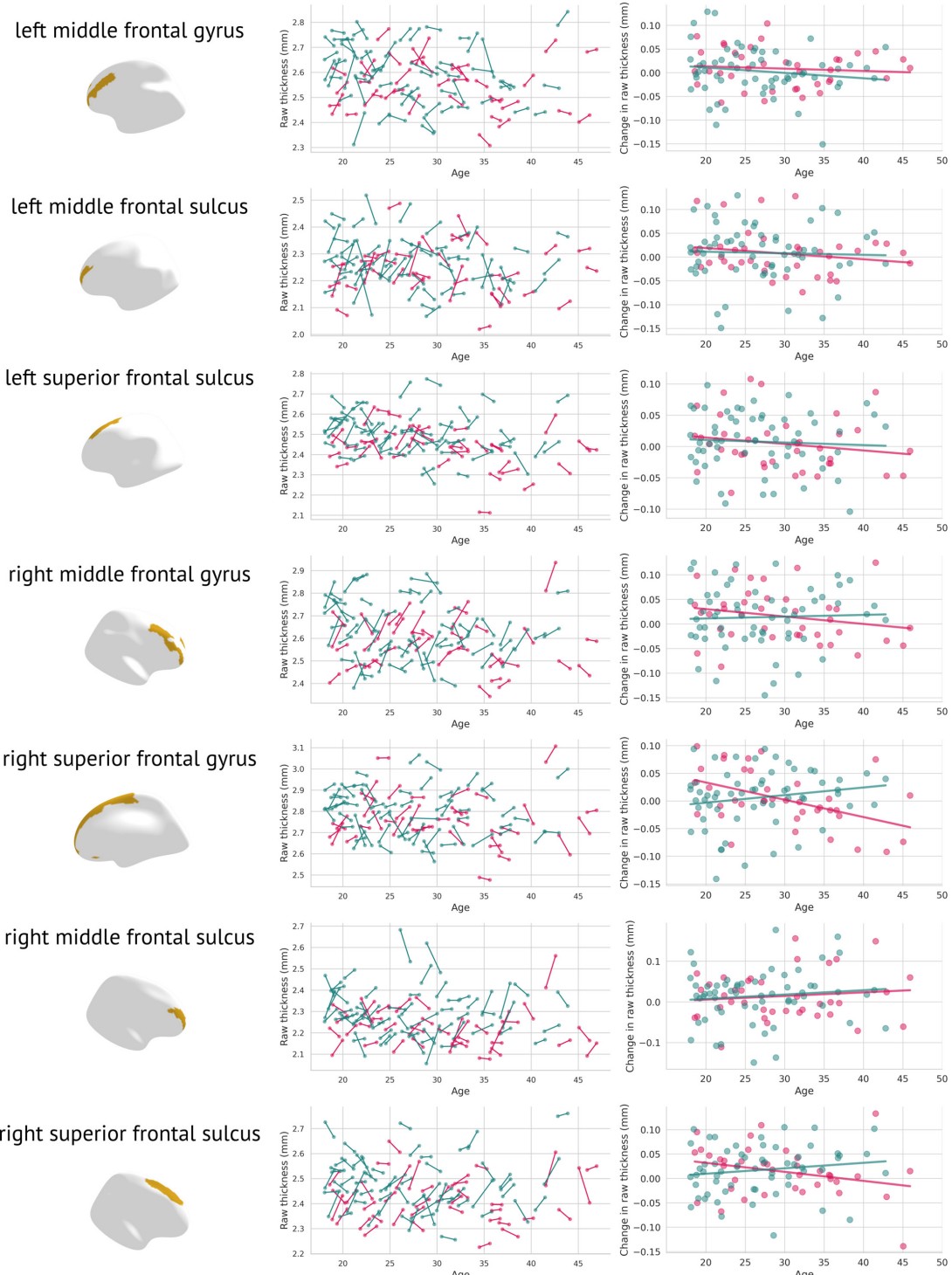

**Appendix 5—figure 3.** Raw changes in grey matter thickness: Each significantly changed region is presented twice, once as a scatter plot containing the original grey matter thickness for both visits (left); females are plotted in pink, males in blue. The figure on the right depicts visit 2 minus visit 1 in raw thicknesses (separately for females—pink, and males—blue).

