## [Editor Report · eLife Assessment]

This paper addresses an **important** topic (normative trajectory modelling), seeking to provide a method aiming to accurately reflect the individual deviation of longitudinal/temporal change compared to the normal temporal change characterized based on a pre-trained population normative model. The evidence provided for the new methods is **solid**.

---

## [Referee Report · Reviewer #2 (Public review)]

Summary:

In this manuscript, the authors provide a method aiming to accurately reflect the individual deviation of longitudinal/temporal change compared to the normal temporal change characterized based on pre-trained population normative model (i.e., a Bayesian linear regression normative model), which was built based on cross-sectional data. This manuscript aims at solving a recently identified problem of using normative models based on cross-sectional data to make inferences about longitudinal change.

Strengths:

The efforts of this work make a good contribution to addressing an important question of normative modeling. With the greater availability of cross-sectional studies for normative modeling than longitudinal studies, and the inappropriateness of making inferences about longitudinal subject-specific changes using these cross-sectional data-based normative models, it's meaningful to try to address this gap from the aspect of methodological development.

---

## [Author Response]

The following is the authors’ response to the previous reviews.

**Public Reviews:**

**Reviewer #2 (Public review):**
Summary:In this manuscript, the authors provide a method aiming to accurately reflect the individual deviation of longitudinal/temporal change compared to the normal temporal change characterized based on pre-trained population normative model (i.e., a Bayesian linear regression normative model), which was built based on cross-sectional data. This manuscript aims at solving a recently identified problem of using normative models based on cross-sectional data to make inferences about longitudinal change.Strengths:The efforts of this work make a good contribution to addressing an important question of normative modeling. With the greater availability of cross-sectional studies for normative modeling than longitudinal studies, and the inappropriateness of making inferences about longitudinal subject-specific changes using these cross-sectional data-based normative models, it's meaningful to try to address this gap from the aspect of methodological development.In the 1st revision, the authors added a simulation study to show how the performance of the classification based on z-diff scores relatively changes with different disruptions (and autocorrelation). Unfortunately, in my view this is insufficient as it only shows how the performance of using z-diff score relatively changes in different scenarios. I would suggest adding the comparison of performance to using the naïve difference in two simple z-scores to first show its better performance, which should also further highlight the inappropriate use of simple z-scores in inferring within-subject longitudinal changes.

Thank you for the suggestion for additional comparison, which we have now implemented in the simulated methods comparison, see Figure 2 and the extended text of Section 2.1.4 Simulation study.

Specifically, we have revised the simulation section to not only illustrate the performance of our z-diff method under various scenarios but also to include a direct comparison with a naïve approach that subtracts two z-scores.

The updated results demonstrate that, compared to the naïve method, the z-diff score consistently maintains a fixed false-positive rate, making it a more robust and controllable approach. Additionally, we show that under conditions of high autocorrelation, the z-diff method is significantly more sensitive in detecting smaller changes than the subtraction method. Importantly, our analysis of a sample from our dataset indicates that high autocorrelation is a prevalent characteristic in real-world data, further supporting the utility of the z-diff method.

We believe that these findings strengthen the case for adopting the z-diff method and underscore the limitations of more intuitive approaches, which, while simple, lack mathematical rigour.

Additionally, Figure 1 is hard to read and obtain the actual values of the performance measure. I would suggest reducing it to several 2-dimensional figures. For example, for several fixed values of rho, how the performance changes with different values of the true disruption (and also adding the comparison to the naïve method (difference in two z-scores)).

We believe that the Reviewer meant Figure 2; indeed, the 3-dimensional visualization, while attractive to some, may have been difficult to read, so we have now replaced it with several 2-dimensional figures as requested.

I would also suggest changing the title to reflect that the evaluation of "intra-subject" longitudinal change is the method's focus.

Thanks for the suggestion. We have now implemented it by changing the title to Using normative models pre-trained on cross-sectional data to evaluate intra-individual longitudinal changes in neuroimaging data.

We hope the changes implemented fulfill the expectations of the Reviewer.